



# Strong variability of the Asian Tropopause Aerosol Layer (ATAL) in August 2016 at the Himalayan foothills

Sreeharsha Hanumanthu[1], Bärbel Vogel[1], Rolf Müller[1], Simone Brunamonti[2], Suvarna Fadnavis[3], Dan Li[1,4], Peter Ölsner[5], Manish Naja[6], Bhupendra Bahadur Singh[3], K. Ravi Kumar[7], Sunil Sonbawne[3], Hannu Jauhiainen[8], Holger Vömel[9], Beiping Luo[2], Teresa Jorge[2], Frank G. Wienhold[2], Ruud Dirkson[5], and Thomas Peter[2]

[1]Institute of Energy and Climate Research (IEK-7), Forschungszentrum Jülich, Jülich, Germany
[2]Institute for Atmospheric and Climate Science (IAC), Swiss Federal Institute of Technology (ETH), Zürich, Switzerland
[3]Indian Institute of Tropical Meteorology (IITM), Pune, India
[4]Key Laboratory of Middle Atmosphere and Global Environment Observation (LAGEO), Institute of Atmospheric Physics, Chinese Academy of Sciences, Beijing, China
[5]Deutscher Wetterdienst (DWD) / GCOS Reference Upper Air Network (GRUAN) Lead Center, Lindenberg, Germany
[6]Aryabhatta Research Institute of Observational Sciences (ARIES), Nainital, India
[7]Centre for Atmospheric Sciences, Indian Institute of Technology (IIT), Delhi, India
[8]Vaisala Oyj, Vantaa, Finland
[9]Earth Observing Laboratory, National Center for Atmospheric Research, Boulder, CO, USA

**Correspondence:** B. Vogel (b.vogel@fz-juelich.de)

**Abstract.**

The South Asian summer monsoon is associated with a large-scale anticyclonic circulation in the Upper Troposphere and Lower Stratosphere (UTLS), which confines the air mass inside. During boreal summer, the confinement of this air mass leads to an accumulation of aerosol between about 13 km and 18 km (360 K and 440 K potential temperature), this accumulation of aerosol constitutes the Asian Tropopause Aerosol Layer (ATAL). We present balloon-borne aerosol backscatter measurements of the ATAL performed by the Compact Optical Backscatter Aerosol Detector (COBALD) instrument in Nainital in Northern India in August 2016, and compare these with COBALD measurements in the post-monsoon time in November 2016. The measurements demonstrate a strong variability of the ATAL's altitude, vertical extent, aerosol backscatter intensity and cirrus cloud occurrence frequency. Such a variability cannot be deduced from climatological means of the ATAL as they are derived from satellite measurements. To explain this observed variability we performed a Lagrangian back-trajectory analysis using the Chemical Lagrangian Model of the Stratosphere (CLaMS). We identify the transport pathways of air parcels contributing to the ATAL over Nainital in August 2016, as well as the source regions of the air masses contributing to the composition of the ATAL. Our analysis reveals a variety of factors contributing to the observed day-to-day variability of the ATAL: continental convection, tropical cyclones (maritime convection), dynamics of the anticyclone and stratospheric intrusions. Thus, the ATAL is a mixture of air masses coming from different atmospheric height layers. In addition, contributions from the model boundary layer originate in different geographic source regions. The location of strongest updraft along the backward trajectories reveal a cluster of strong upward transport at the southern edge of the Himalayan foothills. From the top of the convective outflow level (about 13 km; 360 K) the air parcels ascend slowly to ATAL altitudes within a large-scale upward spiral driven by the





diabatic heating in the anticyclonic flow of the South Asian summer monsoon at UTLS altitudes. Cases with a strong ATAL typically show boundary layer contributions from the Tibetan Plateau, the foothills of the Himalayas and other continental regions below the Asian monsoon. Weaker ATAL cases show higher contributions from the maritime boundary layer, often related to tropical cyclones, indicating a mixing of unpolluted and polluted air masses. Because of the strong growth of Asian

economies, increasing anthropogenic emissions in the future are expected to enhance the thickness and intensity of the ATAL, thereby also enhancing the global stratospheric aerosol loading, which likely impacts surface climate.

*Copyright statement.* Authors

## 1  Introduction

The existence of an Asian Tropopause Aerosol Layer (ATAL) in Northern hemisphere summer (June, July and August) was first

discovered in lidar measurements of aerosol particles in the Upper Troposphere and Lower Stratosphere (UTLS) performed by CALIPSO, the Cloud-Aerosol Lidar and Infrared Pathfinder Satellite Observation (Vernier et al., 2011, 2015). The ATAL occurs between about 360 K and 440 K potential temperature (between about 13 and 17 km); based on CALIOP measurements, the thickness of the ATAL is estimated to be 3–4 km between 30°N and 40°N and is thinner near the equator (Vernier et al., 2011, 2015, 2018; Bian et al., 2020). The existence of the ATAL is also confirmed by in situ balloon-borne backscatter mea-

surements (Vernier et al., 2015, 2018; Brunamonti et al., 2018). Solar occultation observations by the Stratospheric Aerosol and Gas Experiment (SAGE) II (Thomason and Vernier, 2013) indicated the absence of the ATAL prior to 1999, although a recent study suggested that an ATAL already existed in 1997 (Höpfner et al., 2019).

The existence of the ATAL is related to the presence of the Asian summer monsoon circulation and, in particular, to the Asian monsoon anticyclonic circulation at UTLS altitudes from about June to September. The Asian summer monsoon circulation is

affected by the presence of the orography of the Himalayas and the adjacent mountain ranges (Boos and Kuang, 2010). The monsoon anticyclone is linked to deep convection in summer over the Indian subcontinent and the associated diabatic heating (Hoskins and Rodwell, 1995; Randel and Park, 2006). Deep convection occurs primarily just south of and over the lower elevations of the Himalayan barrier; isolated deep convective events occur over the Tibetan Plateau (Houze Jr. et al., 2007). This deep convection in the region of the Asian monsoon anticyclone transports pollutants (e.g., CO, HCN, $C_2H_6$, $SO_2$, $NH_3$,

HCFC-22) emitted at the surface in Asia to UTLS altitudes, where they are confined in the anticyclonic circulation in summer to a certain extent (e.g., Park et al., 2008; Randel et al., 2010; Chen et al., 2012; Ploeger et al., 2015; Chirkov et al., 2016; Garny and Randel, 2016; Santee et al., 2017; Höpfner et al., 2019; Bian et al., 2020).

The source regions of ATAL aerosols and their chemical precursors on the Earth's surface (origin) as well as the transport pathways from the surface to ATAL altitudes are poorly understood. The contributions of different boundary source regions in

Asia to the chemical composition of the Asian monsoon anticyclone are discussed in a variety of studies (e.g. Li et al., 2005; Park et al., 2009; Chen et al., 2012; Bergman et al., 2013; Fadnavis et al., 2014; Vogel et al., 2015; Pan et al., 2016; Li et al.,





2017; Fairlie et al., 2020), identifying sources in India, the Tibetan Plateau, southwest China, southeast Asia, as well as from the Western Pacific. In addition, there is an intraseasonal change of boundary layer source regions contributing to the composition of the anticyclone that depends on the variability of convection and the spatial coverage of the anticyclone itself (Vogel et al., 2015). Further in model simulations, it was found that during the monsoon peak season large quantities of dust are transported

from the Middle East deserts to the upper troposphere and accumulate over the southern and eastern foothills of the Tibetan Plateau (Fadnavis et al., 2013; Lau et al., 2018). Lau et al. (2018) highlighted two key pathways lofting carbonaceous aerosols and dust by orographically forced deep convection, one over the Himalayas-Gangetic Plain (India) and another one over the Sichuan Basin (China). Based on aircraft measurements in summer 2017, Höpfner et al. (2019) concluded that enhanced $NH_3$ concentrations (precursor of solid ammonium nitrate particles) over India were injected by strong convection into the upper

troposphere over northwest India and northeast Pakistan. Source apportionment of the model simulations by Fairlie et al. (2020) indicated the dominance of the contribution of regional anthropogenic emissions from China and the Indian subcontinent to aerosol concentrations in the ATAL in August 2013.

The deep convection in summer over the Indian subcontinent also causes lower concentrations of stratospheric tracers, in particular ozone, in the Asian monsoon anticyclone, whereas water vapour shows a local maximum in the monsoon anticyclone

(e.g., Park et al., 2007; Bian et al., 2012; Randel et al., 2015; Brunamonti et al., 2018). The tropospheric signature inside the anticyclone at potential temperatures between $\sim$360 K and $\sim$440 K (corresponding to about 13 to 18 km) stems from convective transport typically up to the lower end of this range, followed by slow diabatic uplift superimposed on the anticyclonic motion, which Vogel et al. (2019) referred to as "upward spiralling range". Brunamonti et al. (2018) denote the altitude range between maximum convective outflow and the cold point tropopause as the Asian tropopause transition layer (ATTL). Above, the ATTL,

based on $H_2O$ measurements and trajectory calculations, Brunamonti et al. (2018) found a layer of a confined air in the lower stratosphere, which is consistent with the concept of an upward spiralling range.

Particularly strong emissions of $NO_x$, $SO_2$, $NH_3$, and particulate matter occur in Northern India and China (van der A et al., 2017; van Damme et al., 2018; Zheng et al., 2018), which impact ozone and aerosol in the upper troposphere. Ozone and aerosol changes in the upper troposphere in turn have radiative impacts over the region (Roy et al., 2017; Fadnavis et al., 2018;

Bian et al., 2020). However, after the year 2013, because of the implementation of new emission control measures in China, China's anthropogenic emission of some pollutants decreased substantially (e.g., $SO_2$, $NO_x$, CO) while emissions of $NH_3$ and nonmethane volatile organic compounds did not change much during 2010-2017 (Zheng et al., 2018).

The confinement of air masses of tropospheric origin in the monsoon anticyclone also affects the aerosol particles constituting the ATAL (Yu et al., 2017; Vernier et al., 2018; Höpfner et al., 2019). Particles in the ATAL originate from ground

sources (both gas-phase precursors and aerosol particles), which are lifted to UTLS altitudes by convection. The formation mechanism of the aerosol particles constituting the ATAL is not well characterised, but there is evidence that formation of secondary aerosol in the upper troposphere within the Asian monsoon anticyclone at high relative humidity and low temperature is important (Vernier et al., 2015, 2018; Yu et al., 2017).

Only a few measurements of size distributions of particles in the ATAL are available, but measurements with an optical

particle counter (OPC) (Deshler et al., 2003) from Hyderabad, India and measurements with a Printed Optical Particle Spec-





trometer (POPS) (Gao et al., 2016) from Kunming, China, indicate that ATAL particles are dominated by particles with a size of $\approx 0.1\,\mu m$ (Yu et al., 2017; Vernier et al., 2018). Based on CALIOP measurements, the summertime aerosol optical depth associated with the ATAL is $\approx 0.006$ (Vernier et al., 2015), leading to a reduction in insolation of about $0.6\,\mathrm{W/m^2}$ and a surface cooling of roughly $0.5\,\mathrm{K}$.

The impact of the ATAL is also modulated by El Niño. During El Niño, the ATAL is thicker and broader over the Indian region, resulting in a reduction of the solar flux and a surface cooling of about 1 K over North India. An elevated ATAL over South Asia exacerbates the severity of Indian droughts (Fadnavis et al., 2019b). The radiative forcing by the ATAL can further be enhanced by volcanic eruptions in the region; Fairlie et al. (2014, Fig. 8), based on monthly accumulations of CALIPSO aerosol data between 14 and 40 km altitude, reported a top of the atmosphere radiative forcing locally over Asia and Europe
during July 2011 in response to the Nabro volcanic eruption of about $-0.8$ to $-1.5\,\mathrm{W/m^2}$. Vernier et al. (2015) found a positive trend in summertime UTLS aerosol optical depth of about 0.004 resulting in a regional reduction of insolation (total sky radiative forcing) of around $-0.1\,\mathrm{W/m^2}$ over the years 1997–2015, compensating about one third of the radiative forcing associated with the global increase in $CO_2$.

The chemical signature of air masses within the monsoon anticyclone (e.g. tropospheric pollutants, water vapour) is exported
to the Northern Hemisphere during summer and fall through quasi-isentropic transport from low latitudes (e.g., Ploeger et al., 2013; Vogel et al., 2014, 2016, 2019; Spang et al., 2015; Garny and Randel, 2016; Müller et al., 2016; Fadnavis et al., 2018; Rolf et al., 2018; Yan et al., 2019). Because of this export of air from the Asian monsoon circulation to the Northern Hemisphere, the ATAL particles contribute significantly ($\sim 15\%$) to the Northern Hemisphere stratospheric column aerosol surface area on an annual basis (Yu et al., 2017).

Only limited information on the chemical composition of the ATAL particles is available from measurements so far. From simulations, there is evidence that desert dust is lifted to UTLS altitudes and entrained into the ATAL (Fadnavis et al., 2013; Lau et al., 2018; Yuan et al., 2019). Aircraft in situ measurements suggest that at lower altitudes the chemical composition of ATAL particles are dominated by carbonaceous and sulphate materials, consistent with the expectation that aerosol trends in the UTLS in the past decades are under increasing influence of sulphur emissions in Asia (Martinsson et al., 2014; Vernier et al.,
2015; Fadnavis et al., 2019a). However, the first offline (balloon-borne filter samples) chemical analysis of ATAL particles suggested the presence of nitrate aerosol, but undetectable concentrations of sulphate ions (Vernier et al., 2018). Also, the evaluation of a set of remote sensing measurements indicates a strong contribution of solid ammonium nitrate particles to the ATAL (Höpfner et al., 2019). Moreover, Fairlie et al. (2020) found in model simulations for summer 2013 a different chemical compositions of the ATAL depending on the location within the Asian monsoon anticyclone. They found that nitrate aerosol is
a dominant component of the ATAL on the southern flank of the anticyclone.

The Asian monsoon in the UTLS does not constitute a stable, persistent unimodal anticyclonic circulation (as it sometimes appears in climatologies), but can be bimodal (with an Iranian and a Tibetan mode) and moreover shows a strong day-to-day variability (Zhang et al., 2002; Yan et al., 2011; Vogel et al., 2015; Nützel et al., 2016). There is also a substantial inter-annual variability of the monsoon circulation with impact on the concentrations of tracers confined in the monsoon circulation (Santee





et al., 2017; Yuan et al., 2019). This variability of the monsoon circulation impacts on the variability of trace-gas (Luo et al., 2018) and aerosol distributions (Lau et al., 2018; Yuan et al., 2019).

The strong convective activity in the Asian summer monsoon region also impacts the temperature in the monsoon region and the formation of cirrus clouds. Enhanced convection is linked to cold anomalies in the subtropical lower stratosphere

(Park et al., 2007; Randel et al., 2015). Further, the occurrence fractions of cirrus in the middle to upper (16–18 km) tropical tropopause layer (TTL) (e.g., Fueglistaler et al., 2009) exhibit a pronounced maximum over the Asian monsoon region, both in observations and in model simulations (Ueyama et al., 2018). Convection is likely the dominant driver of localised upper tropospheric $H_2O$ and cloud maxima in the region (Park et al., 2007; Ueyama et al., 2018). Ice clouds in the tropical deep convective regions are important as they exert a considerable net warming effect (Hong et al., 2016).

To address fundamental open questions and uncertainties regarding the anticyclone and the ATAL, a number of balloon-borne campaigns have been conducted in summer in the Asian monsoon region employing the Compact Optical Backscatter Aerosol Detector (COBALD) instrument (Bian et al., 2012, 2020; Vernier et al., 2018; Brunamonti et al., 2018). Here we present an analysis of the COBALD measurements in Nainital in Northern India in August and November 2016. The signature of the ATAL is clearly visible in most of the soundings in August (but not in every single one), but we find a substantial variability of

the ATAL, both in backscatter intensity and in altitude range of the ATAL. A Lagrangian back-trajectory analysis is performed in order to identify the air mass origin in the model boundary layer and the transport pathways of air parcels contributing to the ATAL over Nainital in August 2016. The paper is organised as follows: Sect. 2 describes the soundings in Nainital and the CLaMS trajectory calculations, Sect. 3 presents the results of aerosol backscatter measurements, Sect. 4 discusses detailed trajectory calculations for three selected days as well as the simulation results for all balloon flight from Nainital. Finally, our

results are discussed in Sect. 5 and the conclusions are given in Sect. 6.

## 2 Methods

The findings presented here are based on a set of balloon measurements performed in Nainital, Uttarakhand, India (29.35°N, 79.46°E); Nainital is located at an elevation of 1820 m above sea level. The Aryabhatta Research Institute of Observational

Sciences (ARIES) provided the campaign base to perform the launches. These measurements were conducted in August and November 2016 and are described in detail by Brunamonti et al. (2018, 2019). A brief summary is given here.

### 2.1 Balloon-borne instrumentation

The aerosol and cirrus measurements analysed here are based on balloon-borne backscatter measurements by the Compact Optical Backscatter Aerosol Detector (COBALD). COBALD operates at optical wavelengths of 455 nm (blue visible) and

940 nm (infrared) and was developed at the ETH Zürich (e.g. Wienhold, 2008; Brabec et al., 2012). It was hosted by a standard RS41-SGP meteorological radiosonde and the DigiCORA MW41 sounding system (Vaisala, Finland). In addition, the balloon payload included an electrochemical concentration cell (ECC, manufacturer: EN-SCI, USA) (Komhyr, 1969; Komhyr et al.,





1995) for measuring the ozone mixing ratio and a cryogenic frost-point hygrometer (CFH, EN-SCI, USA) (Vömel et al., 2007, 2016) for measuring the water vapour mixing ratio (Brunamonti et al., 2019; Jorge et al., 2020). We use the pressure measured by the RS41 as the main vertical coordinate for all instruments. The ATAL analysis is mainly based on the COBALD 455 nm measurements, in particular on the backscatter ratio (BSR), which is defined as the ratio of the COBALD raw signal (from

particulates and air molecules) over the pure molecular scattering (derived from the ambient molecular number density, using the temperature and pressure measured by RS41). The colour index (CI), defined as the 940-to-455 nm ratio of the aerosol component of the BSR, i.e. $CI = (BSR_{940} - 1)/(BSR_{455} - 1)$ is a useful indicator of particle size and can thus be employed to discriminate aerosol and cirrus measurements (Cirisan et al., 2014; Vernier et al., 2015; Brunamonti et al., 2018). The CI for all balloon flights over Nainital in August 2016 is shown in Fig. A1 in Appendix A. For the detection of aerosol particles, we

consider the short wavelength channel (455 nm), which is more sensitive to smaller particles than the long wavelength channel (940 nm). It also provides a smaller value for the maximum uncertainty (1% and 5% for the shortwave and longwave channel, respectively) and is thus less noisy (Brabec et al., 2012) (see Appendix B for details).

## 2.2    Campaign description

From Nainital, at the foothills of the Himalayas, 15 successful night-time balloon launches were conducted during the peak of

the monsoon season in August 2016 (Tab. 1). All balloon soundings with a COBALD instrument were launched at night-time between 23:00 Indian Standard Time (IST) (corresponding to Coordinated Universal Time (UTC) +6 h) and next day 03:00 IST. Due to the high sensitivity of its photodiode detector, COBALD can only be deployed during night-time. The frequency of soundings was adjusted depending on the meteorological conditions. Out of a total of 30 balloon soundings (Brunamonti et al., 2018), we present here 17 launches (15 in August, 2 in November), which were performed with the COBALD instrument

(Tab. 1). Four launches were excluded in our analysis. The 16 August launch burst below 3 km altitude due to harsh weather conditions, and on 28 August the data showed very low signal-to-noise ratio in the stratosphere, likely caused by optical contamination. The post processing of the COBALD data requires the normalisation of the raw signal to a reference value, which is typically based on the measurements from above 30 km altitude (i.e. above the stratospheric aerosol layer). Due to the low absolute BSR signal of the ATAL, a careful calibration of all profiles is critical here. Since no stratospheric data are

available on 5 and 12 August due to early bursts (below 20 km altitude), for the sake of consistency, we exclude these two profiles from our analysis.

## 2.3    Data analysis and processing

In order to identify the ATAL, it is necessary to discriminate aerosol and cirrus clouds (ice particles) in the backscatter measurements. For this purpose, we used the ice saturation ($S_{ice}$) from the CFH instrument and the CI from the COBALD measurements

(Cirisan et al., 2014). We reject layers with $CI > 7.0$, $BSR_{940} \geq 2$ and $S_{ice} > 70\%$ as cirrus clouds (Vernier et al., 2015; Li et al., 2018; Brunamonti et al., 2018). Other sections of the profiles measured during the August soundings, which show substantially elevated values of $BSR_{455}$ in the UTLS, are classified as ATAL. As we will see below, the classification is considerably simplified by the fact that the COBALD profiles reveal clearly the fingerprints of the ATAL, i.e. top and bottom of the aerosol



| No. | Date | Pressure [hPa] | pot. Temperature [K] | $\overline{BSR}_{455}$ | $\sigma$ | note |
|-----|------|----------------|----------------------|-----------------------|----------|------|
| NT001 | 02-08-16 | 141 – 82 | 365 – 405 | 1.067 | 0.008 | |
| NT002 | 03-08-16 | 122 – 82 | 368 – 400 | 1.092 | 0.013 | bottom Cirrus |
| NT003 | 05-08-16 | 115 – 78 | 374 – 415 | 1.067 | 0.010 | |
| NT004 | 06-08-16 | 135 – 95 | 364 – 388 | 1.083 | 0.009 | |
| NT005 | 08-08-16 | 156 – 82 | 362 – 408 | 1.071 | 0.005 | |
| NT007 | 11-08-16 | 105 – 88 | 367 – 400 | 1.070 | 0.024 | bottom Cirrus |
| NT009 | 12-08-16 | 140 – 92 | 363 – 385 | — | — | Cirrus |
| NT011 | 15-08-16 | 140 – 92 | 366 – 389 | 1.023 | 0.009 | no ATAL |
| NT015 | 17-08-16 | 150 – 75 | 363 – 422 | 1.076 | 0.014 | |
| NT017 | 18-08-16 | 150 – 75 | 362 – 422 | 1.065 | 0.008 | |
| NT018 | 19-08-16 | 128 – 73 | 365 – 420 | 1.073 | 0.010 | bottom Cirrus |
| NT023 | 21-08-16 | 115 – 80 | 372 – 404 | 1.056 | 0.011 | |
| NT025 | 23-08-16 | 120 – 85 | 369 – 398 | 1.054 | 0.007 | |
| NT027 | 26-08-16 | 95 – 83 | 382 – 398 | 1.080 | 0.004 | bottom Cirrus |
| NT029 | 30-08-16 | 115 – 74 | 370 – 418 | 1.059 | 0.006 | |
| NT033 | 10-11-16 | 140 – 92 | 365 – 394 | 1.037 | 0.006 | post-monsoon |
| NT034 | 11-11-16 | 140 – 92 | 359 – 401 | 1.034 | 0.003 | post-monsoon |

**Table 1.** Table of all night-time balloon launches analysed here. Listed are number, date, top and bottom of the ATAL noted in both pressure and potential temperature, the mean value of the backscatter ratio ($\overline{BSR}_{455}$ excluding cirrus contributions) and its standard deviation ($\sigma$) (see Sects. 2.2 and 3.1 for details). Top of the ATAL is defined as the pressure level, where either the $BSR_{455}$ profile has the strongest gradient or where $BSR_{455}$ merge to the November background. The bottom is defined using the same criteria, except for cases where ATAL is limited to below by a cirrus cloud (3, 11, 19 and 26 August). On 12 August, the UTLS is filled by a 5 km thick cirrus cloud, hence the ATAL cannot be diagnosed. On 15 August 2016, the UTLS was largely cloud-free but without any indication of the ATAL. The November soundings are post-monsoon without the ATAL. For soundings without the ATAL a reference pressure (140–92 hPa) and potential temperature (calculated from the measured temperature and pressure) range is given. This reference range allows comparisons with the ATAL observations to be made (details see Sect. 4).





layer can be identified with reasonable precision. We quantify the elevated values of $BSR_{455}$ by comparing the August profiles with the mean of the November measurements, when there is no ATAL during post-monsoon (Brunamonti et al., 2018). The enhancement of $BSR_{455}$ for the ATAL remains below 1.12 (and the CI below 7). Conditions with cirrus clouds embedded within the ATAL can also be easily identified, as the cirrus clouds have 10–100 times larger $BSR_{455}$. As they completely mask

these ATAL particles, these cannot be detected and quantified under such conditions (see Section 3 below).

Following earlier work (Brabec et al., 2012; Brunamonti et al., 2018), we use here vertically binned data for the CFH and COBALD instrument. $BSR_{455}$, $BSR_{940}$, CI and $S_{ice}$ are binned in pressure intervals of 1 hPa for pressure greater than 300 hPa and 0.5 hPa for pressure less than 300 hPa, which yields an improved signal to noise ratio. This binning corresponds to a vertical resolution of approximately 25 m in the UTLS. All data were carefully quality checked and measurements showing evidence

of anomalous instrumental behaviour were rejected, as described by Brunamonti et al. (2018). The processing of COBALD includes the rejection of 'moon spikes' that may arise due to the oscillatory motion of the payload 60 m below the balloon, when the detector happens to be pointing towards the moon. Moon spikes affect only a tiny fraction of the COBALD data and care is taken not to confuse them with thin cirrus clouds (details see Appendix C).

## 2.4 Trajectory calculations

For the model analysis, we employ the trajectory module of the three-dimensional Lagrangian chemistry transport model CLaMS (McKenna et al., 2002; Pommrich et al., 2014) driven by dynamic fields from the European Centre of Medium-Range Weather Forecasts (ECMWF) interim reanalysis (ERA-Interim) employing a horizontal resolution of $1° \times 1°$ (Dee et al., 2011). In ERA-Interim changes are implemented to improve deep and mid-level convection compared to previous reanalysis data (Dee et al., 2011). CLaMS back trajectory calculations are very well suited to analyse the detailed transport pathway of an air parcel,

although they consider only the advective transport, neglecting mixing processes entirely. Earlier, CLaMS trajectories were applied to a variety of problems such as polar chlorine chemistry as well as transport in the tropics, in particularly in the region of the Asian monsoon and in tropical cyclones (e.g., Vogel et al., 2003; Ploeger et al., 2010, 2012; Vogel et al., 2014, 2019; Li et al., 2017, 2018, 2020). To analyse the transport pathways and the origin of air masses contributing to the ATAL in August 2016 over Nainital, CLaMS 40-day backward trajectory calculations were performed. As the starting point for

backward trajectory calculation we use the pressure levels of the measurements recorded every second (i.e. no binned data). The same trajectory model set-up is used as in several previous publications (Vogel et al., 2014; Li et al., 2017, 2018, 2020). In the CLaMS model, potential temperature is used as the vertical coordinate when the pressure is less than about 300 hPa, (i.e. in the upper troposphere and in the stratosphere); when the pressure is greater than about 300 hPa (more accurately, for pressure $p$ exceeding a reference level of $p/p_{surface} = 0.3$) a pressure-based orography-following hybrid coordinate $\zeta$ (in units of K) is

used (Ploeger et al., 2010; Pommrich et al., 2014). When the back trajectory dives into the CLaMS model boundary layer, we consider the air mass origin to be reached. The model boundary layer is defined as the hybrid pressure/potential temperature coordinate $\zeta < 120$ K (Pommrich et al., 2014; Vogel et al., 2019), which corresponds to a layer $\approx$ 2–3 km above the surface of the Earth considering orography.





## 3 The observations in Nainital in 2016

### 3.1 Aerosol backscatter measurements in August 2016

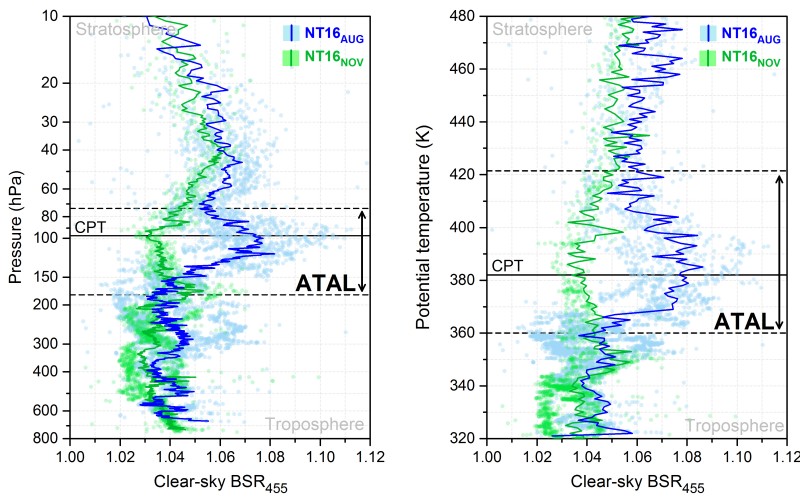

**Figure 1.** Backscatter measurements (BSR$_{455}$) in Nainital, India, in August 2016 versus pressure (left) and versus potential temperature (right). Shown are BSR$_{455}$ values for aerosol (dots) and averaged aerosol profiles (solid lines), both without cirrus contributions for August (blue) and November (green). Black lines show the mean temperature lapse-rate minimum (lower dashed), cold point tropopause (CPT, solid) and top of confinement (upper dashed). The black vertical arrows indicate the the ATAL region (Left panel: adapted from Fig. 11 of Brunamonti et al. (2018); Right panel: the ATAL boundaries and CPT from Tab. 3 in Brunamonti et al. (2018)).

An overview of the Nainital aerosol backscatter observations in August (blue) and November (green) 2016 with pressure as the vertical coordinate is provided in Fig. 1 (left). Like for other years and locations (Vernier et al., 2015, 2018), the ATAL is clearly visible as an enhancement in backscatter ratio (BSR$_{455}$) in August compared to the November measurements, with the averaged backscatter (BSR$_{455}$) values in August reaching about 1.08 (Fig. 1). In Brunamonti et al. (2018), the ATAL is defined between the mean potential temperature lapse-rate minimum and the top of the confined lower stratosphere (Fig. 1, dashed lines). The potential temperature lapse-rate minimum coincides approximately with the top of the convective outflow level ($\approx$180 hPa) (e.g., Gettelman and de Forster, 2002; Brunamonti et al., 2018).

Fig. 1 (right) shows the same BSR$_{455}$ values, but with potential temperature as the vertical coordinate. Using the backscatter measurements with potential temperature as the altitude scale, a somewhat different picture emerges than when using pressure as the altitude scale. In particular, a sharp increase in the averaged BSR$_{455}$ in August around 360–365 K is clearly noticeable. This steep gradient represents the top of the convective outflow region (e.g., Gettelman and de Forster, 2002), i.e. below this level more frequent deep convection scavenges the aerosol. An enhancement of the averaged BSR$_{455}$ in August compared to November is noticeable up to potential temperatures of about 420 K ($\sim$75 hPa), but especially when considering potential





temperature as the vertical coordinate, a top of the ATAL is not clearly defined. Only a slow decrease of averaged $BSR_{455}$ with altitude is observed (Fig. 1, right). This observation is consistent with a decreasing confinement of the air mass within the Asian monsoon anticyclone with increasing potential temperature (Brunamonti et al., 2018; Vogel et al., 2019).

In Fig. 2, all 15 investigated COBALD balloon profiles in August 2016 (see also Tab. 1) are shown individually (blue lines) in
comparison to the average of the two November measurements (green lines) used as a background signal. The top and bottom of the ATAL was determined for each balloon profile individually (quantified in Tab. 1) and these levels are shown in Fig. 2 as horizontal black dashed lines. Here we approximate the top and bottom of the ATAL by the pressure level where the $BSR_{455}$ profile has the strongest gradients or where the $BSR_{455}$ merges with the November background. The bottom is set using the same criteria as for the top of the ATAL, except for cases where a cirrus cloud was detected directly below the ATAL. In these
cases the top of the cirrus cloud is set to the bottom of the ATAL which is hidden by the cirrus cloud (3, 11, 19, 26 August).

The mean value of the backscatter intensity ($\overline{BSR}_{455}$) between bottom and top is individually calculated for each balloon sounding using the binned data (see Tab. 1). Cirrus clouds between top and bottom are excluded in the calculation of $\overline{BSR}_{455}$ and its standard deviation. Therefore, no $\overline{BSR}_{455}$ value is shown in Tab. 1 for the 'no ATAL flight' on 12 August, because over the entire altitude range where the ATAL is expected cirrus was detected.

## 3.2   Day-to-day variability of the ATAL

The enhancement of $BSR_{455}$ for the individual measurements in August compared to the mean of the two November measurements is shown in Fig. 2 (orange shading). There is a strong variability of the altitude, the vertical extent and the $BSR_{455}$ intensity within the ATAL. There are even days, when no ATAL is observed over Nainital (12 and 15 August 2016). In addition,
on many balloon flights, a cirrus cloud is observed (e.g. as a very strong enhancement of $BSR_{455}$, see also section 2.3 above); cirrus clouds are marked in grey in Fig. 2. These cirrus clouds can occur below the ATAL, but there are also cases when cirrus clouds (ice particles) of a moderate vertical extent ($\sim$10 K) are measured within the ATAL (17, 18, and 21 August 2016). From the measurements alone we can not exclude that the ATAL and cirrus clouds can coexist in this region (see Sect. 2.3 and discussion in Sect. 5).

From a seasonal average perspective, the ATAL extends in the longitudinal direction from the Middle East to East Asia and meridionally between 15°-45°N (e.g., Vernier et al., 2015; Fairlie et al., 2020). This climatological picture is also true for the Asian monsoon anticyclone itself (e.g., Park et al., 2008; Vogel et al., 2016). Likewise, for the Asian monsoon anticyclone, there is a large variability from day to day in spatial extent, strength, and location, manifesting in an oscillation between a state with one anticyclone and two separated anticyclones (two modes) (e.g., Zhang et al., 2002; Yan et al., 2011; Vogel et al., 2015;
Nützel et al., 2016).

In the following, we consider three specific days more closely, which are examples for particular cases (see also Sec. 4). We consider the case of an established ATAL (6 August 2016), a case, where no ATAL was observed (15 August 2016) and a case influenced by a typhoon (18 August 2016). We selected the 6 August because the mean value of the backscatter intensity

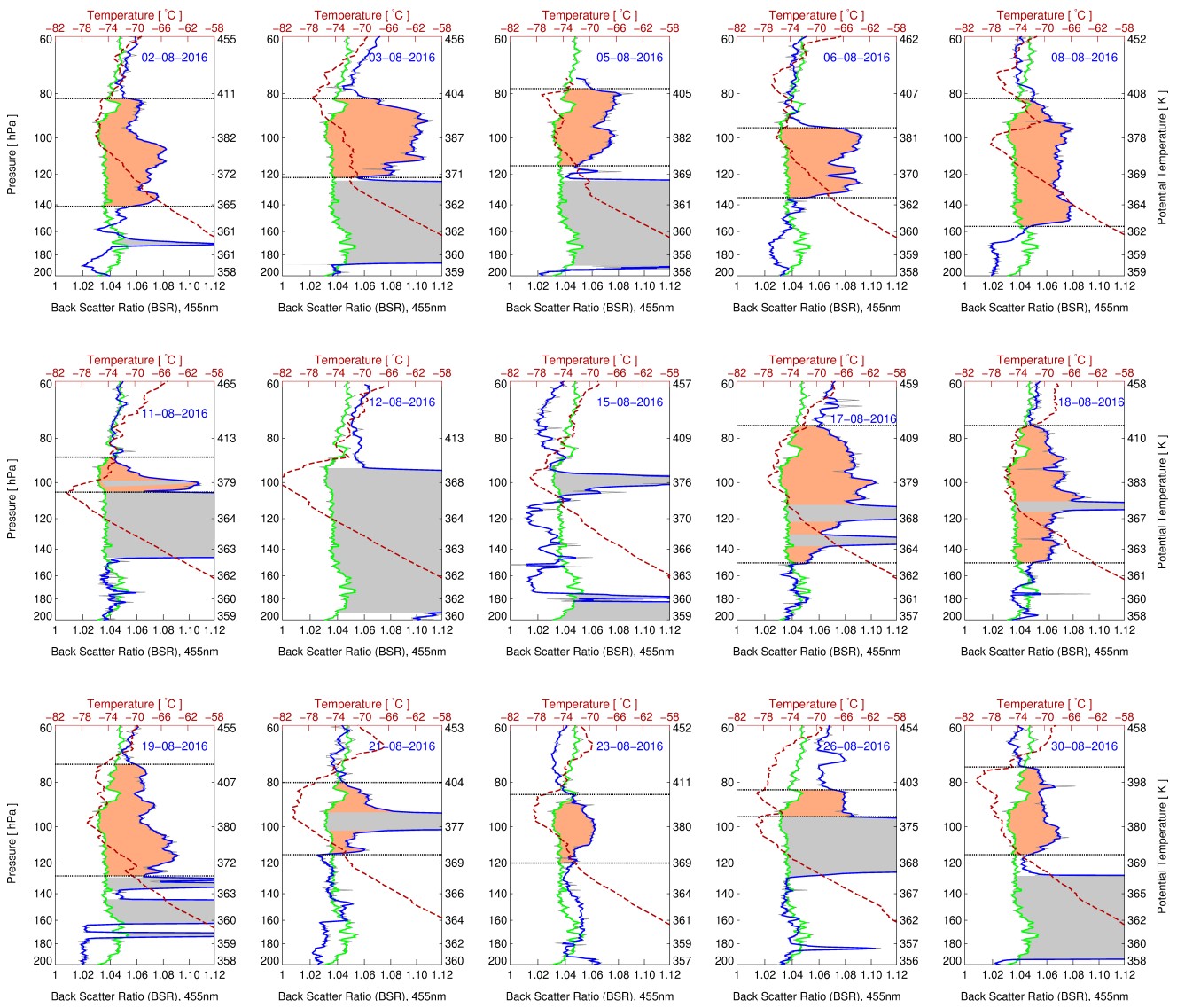

**Figure 2.** COBALD BSR$_{455}$ measurements in the UTLS region versus pressure 60–200 hPa for each flight in August 2016 (blue line). Corresponding potential temperature calculated using both measured pressure and temperature is also given for orientation (see right-hand y-axis). The averaged BSR$_{455}$ profile of the two November measurements is shown in green. The blue and the green lines show a slightly smoothed representation (three point smoothing) of the original 0.5 hPa binned data (light grey lines; see Sect. 2.3 for details). The measured temperature for each day in August 2016 is shown in red. The ATAL is highlighted by the orange shading and cirrus clouds are indicated by the grey shading. The top and bottom boundaries of the ATAL (see Tab. 1) are marked by horizontal dashed black lines, except for 12 and 15 August, as no ATAL was detected on these days.

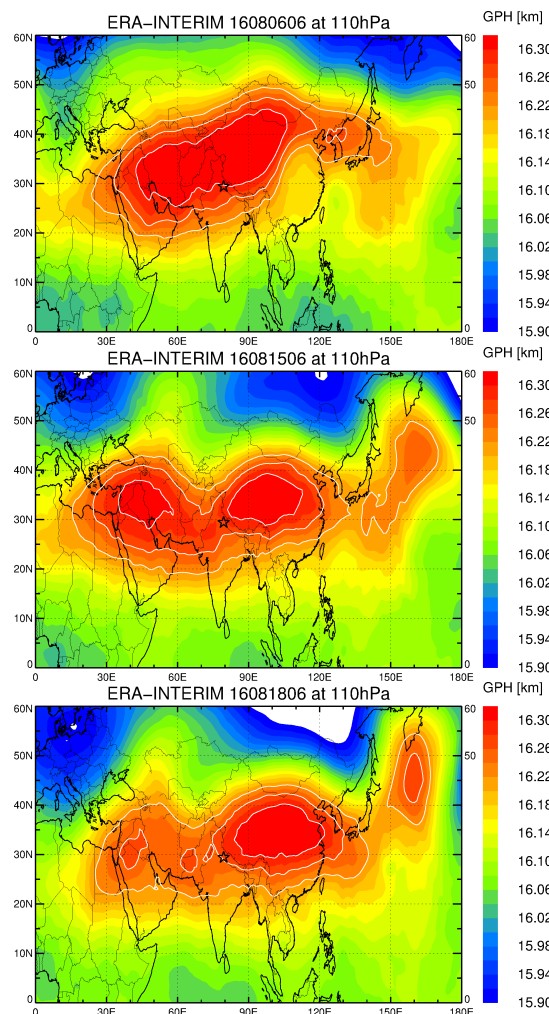

**Figure 3.** Location of the Asian monsoon anticyclone measured by the geopotential height at 110 hPa for 6 August, 15 August and 18 August 2016 at 06:00 UTC (corresponding to 02:30 local time (IST) in Nainital). The GPH of 16.22, 16.26, and 16.30 km are shown as white contour lines. The location of Nainital is marked by a black star symbol.





($\overline{\mathrm{BSR}}_{455}$=1.083) is the highest from all ATAL cases and is not influenced by cirrus. From the two no ATAL cases (12 and 15 August), 15 August is chosen because the cirrus cloud is thinner than on 12 August.

The measurement in Nainital on 18 August 2016 was partially influenced by air uplifted in a tropical typhoon (this case will be discussed in detail in Sec. 4.3). On 18 August the ATAL occurs in an broader altitude range compared to 6 August, however

the intensity of the ATAL is in general lower ($\overline{\mathrm{BSR}}_{455}$=1.065) than on 6 August.

The location of the Asian monsoon anticyclone for these three days is shown in Fig. 3. The Asian monsoon anticyclone is indicated here by geopotential height at 110 hPa (GPH = $\Phi g^{-1}$ with geopotential $\Phi$ [m$^2$ s$^{-2}$] and $g = 9.81$ [m s$^{-2}$]) using ERA-Interim reanalyses for 6, 15, and 18 August 2016. The 110 hPa pressure level is selected because this level is located within the ATAL for most balloon flights in August 2016.

For the case of an established ATAL (6 August 2016) a single-modal anticyclone is found. For the case with no observation of an ATAL (15 August 2016), a bi-modal structure of the anticyclone is found and Nainital is located between the eastern and western mode of the Asian monsoon anticyclone. For the ATAL case impacted by a tropical cyclone (18 August 2016) a single-modal anticyclone is found with pronounced eddy shedding events (outflow from the anticyclone) towards both the east and the west (e.g. Vogel et al., 2015, 2016).

## 4    Results of back-trajectory analysis

As shown in Fig. 2, there is a strong day-to-day variability in the ATAL altitude range, vertical extent and the BSR$_{455}$ intensity. To analyse the transport pathway and the origin of the air masses contributing to the ATAL, for each sounding in August 2016 backward trajectories were calculated over a time period of 40 days. The dependence of the results of the trajectory calculations

on the trajectory length (40, 60, and 80 days) will be discussed in detail below (Sec. 4.4).

The backward trajectories were started every second along the measured balloon profile where an ATAL was detected during the flight (see Tab. 1 and Fig. 2, orange). The back-trajectory calculations are based on 1-sec data having a higher vertical resolution than the BSR$_{455}$ values in Fig. 2 (binned data in pressure intervals of 0.5 hPa). In cases with a thin cirrus cloud embedded in the ATAL (17, 18, and 21 August 2016), the region of the cirrus cloud is included in the backward trajectory

calculation because it is likely that ATAL and ice particles coexisted in this region (see Sect. 2.3 and discussion in Sect. 5). For the two no ATAL cases (12 and 15 August 2016) as well as both post-monsoon measurements (10 and 11 November 2016), backward trajectories were started in a pressure range between 140 and 92 hPa, to allow comparison with the ATAL observations.

We classify the origin of the air masses found in the ATAL according to sources in the model boundary layer, in the lower

troposphere (LT), in the upper troposphere (UT) and in the lower stratosphere (LS) (Tab. 2). Trajectories are considered ending in the model boundary layer (BL) when they are located for the first time below about 2–3 km (i.e., hybrid coordinate $\zeta \leq$ 120 K). The location of this point is referred to as 'end point' of the trajectory in the model boundary layer. The trajectory length for these trajectories ending in the BL is shorter than 40 days because they reach the BL before 40 days. Within 40 days





| No. | Date | BL %<br>$\zeta \leq 120$ K | LT %<br>$\zeta > 120$ K<br>$\theta \leq 340$ K | UT %<br>$\zeta > 120$ K<br>340 K $< \theta \leq 370$ K | LS %<br>$\zeta > 120$ K<br>$\theta > 370$ K | Num. Traj. |
|---|---|---|---|---|---|---|
| NT001 | 02-08-16 | 47.0 | 10.4 | 30.3 | 12.3 | 670 |
| NT002 | 03-08-16 | 39.1 | 17.2 | 36.5 | 7.2 | 419 |
| NT003 | 05-08-16 | 22.3 | 13.8 | 44.6 | 19.4 | 413 |
| NT004 | **06-08-16** | 64.2 | 14.0 | 21.6 | 0.3 | 385 |
| NT005 | 08-06-16 | 54.3 | 15.4 | 30.3 | 0.0 | 680 |
| NT007 | 11-08-16 | 40.9 | 11.2 | 24.9 | 23.0 | 269 |
| NT009 | 12-08-16 | 28.7 | 11.0 | 54.6 | 5.6 | 463 |
| NT011 | **15-08-16** | 44.8 | 19.8 | 30.2 | 5.2 | 444 |
| NT015 | 17-08-16 | 31.8 | 11.1 | 41.6 | 15.5 | 651 |
| NT017 | **18-08-16** | 45.1 | 10.6 | 29.5 | 14.8 | 705 |
| NT018 | 19-08-16 | 27.1 | 10.5 | 29.9 | 32.5 | 569 |
| NT023 | 21-08-16 | 29.6 | 19.6 | 24.8 | 26.0 | 250 |
| NT025 | 23-08-16 | 63.7 | 11.5 | 18.7 | 6.0 | 331 |
| NT027 | 26-08-16 | 13.3 | 16.7 | 50.8 | 19.2 | 120 |
| NT029 | 30-08-16 | 29.6 | 17.3 | 34.7 | 18.4 | 392 |
| NT033 | 10-11-16 | 46.1 | 35.9 | 17.5 | 0.6 | 538 |
| NT034 | 11-11-16 | 13.9 | 22.4 | 57.8 | 6.0 | 519 |

**Table 2.** Contributions of the origin of the air masses found in the ATAL using 40-day backward trajectories according to sources in the boundary layer (BL), in the lower troposphere (LT), in upper troposphere (UT) and in the lower stratosphere (LS). The number of backward trajectories (starting within the ATAL layer) calculated along each balloon profile is given in the last column. The dates of three balloon flights representing typical situations encountered during the measurements in August 2016 are marked in bold type: the case of an established ATAL (6 August 2016), a case without the observation of an ATAL (15 August 2016), and a case, where the measurements are heavily influenced by air masses uplifted in a tropical cyclone (18 August 2016).





$\approx$ 14–64% of the trajectories are reaching the model boundary layer. For the remaining trajectories ending at atmospheric altitudes, a potential temperature criterion is employed to discriminate between origins in the lower troposphere (LT), in the upper troposphere (UT) and in the lower stratosphere (LS) (see Tab. 2).

There is a strong variability of the contributions of different atmospheric height layers (BL, LT, UT and LS) to the air masses in the ATAL (Tab. 2), demonstrating that within the ATAL a mixture of air masses of different origin exist. Before we discuss the results of the back-trajectory calculations for all balloon-flights in Nainital, three days representing typical situations encountered during the measurements in August 2016 are presented in detail: the case of an established ATAL (6 August 2016), a case without the observation of an ATAL (15 August 2016), and a case, where the measurements are heavily influenced by air masses uplifted in a tropical cyclone (18 August 2016).

## 4.1 Case 1: Established ATAL on 6 August 2016

On 6 August 2016 the ATAL was observed between 364 K to 388 K (135−95 hPa). The back-trajectory analysis of 40 days (Tab. 2) shows that 64% of the trajectories are from the boundary layer (BL), 14 % from the lower troposphere (LT), 22 % from the upper troposphere (UT) and only 0.3 % from the lower stratosphere (LS). The 40-day backward trajectories (potential temperature versus time) classified by the location of the trajectory end points at different atmospheric height layers are shown in Fig. 4.

Air masses from the BL are mainly uplifted very fast by individual convection events within short time periods of 1−2 days up to $\approx$ 360 K. Subsequently, slower updraft above 360 K occurs within the Asian monsoon anticyclone (at longitudes between 0°E and 140°E) where the air masses follow the anticyclonic flow (see Fig. 5). This region is referred to 'upward spiralling range' by Vogel et al. (2019). Trajectories from the LT are very similar to those from the BL, however there are more trajectories originating outside of the Asian monsoon region from western longitudes. Most trajectories from the UT experienced radiatively driven updraft within the upward spiralling range. In contrast, the trajectory originating in the LS descends in the stratosphere (coming from Northern America) and subsequently ascent occurs within the anticyclone.

The trajectories originating from the BL are from the region of the Asian monsoon anticyclone and from the western Pacific (Fig. 5). To obtain a deeper insight into the origin of air masses contributing to the ATAL, trajectories originating in the BL are further separated. In a latitude-longitude box from 60°E to 160°E and from 5°S to 45°N (Fig. 6, Tab. 3), we distinguish between Tibetan Plateau (Tibet), foothills (Foothills) including the Himalayan foothills, the remaining continental (Land) and maritime area (Ocean). The continental areas were separated according to geopotential ($\Phi$) or GPH ($\Phi g^{-1}$). The Tibetan Plateau is defined as the altitude above a geopotential $\Phi$ of 40000 [m$^2$ s$^{-2}$] (above GPH $\approx$ 4000 m), the foothills as 20000 < $\Phi$ < 40000 [m$^2$ s$^{-2}$] (GPH $\approx$ 2000−4000 m), the remaining continental area as $\Phi$ < 20000 [m$^2$ s$^{-2}$] (below GPH $\approx$ 2000 m). Trajectories originating outside the latitude-longitude box are classified as Residual.

Fig. 6a shows the location of the end points of all trajectories ending in the BL for 6 August 2016 colour-coded by the transport time from the boundary layer to the location of the measurements. The shortest transport times of about 15−20 days are found for trajectories originating on the Tibetan Plateau. In addition there is a cluster of trajectories from northeast India with transport times of about 25 days.





In addition, the location of the maximum updraft within 18 hours along these 40-day backward trajectories is calculated. It is first determined where the maximum change in potential temperature occurs ($\Delta\Theta_{max}$) along the 40-day backward trajectories (calculated as running mean over the change in potential temperature within 18 hours). The location of the maximum updraft within the 18 hours is then calculated as the mean location of the trajectory within 18 hours for $\Delta\Theta_{max}$ (Fig. 6b).

It is evident that the location of the end points (Fig. 6a) and of the strongest updraft (Fig. 6b) differ substantially. Our calculations show a cluster of locations with an updraft larger then $25\,\mathrm{K\,18h^{-1}}$ within a belt along the southern edge of the Himalayas and over Myanmar. The area of the foothills (GPH $\approx 2000-4000\,\mathrm{m}$) is very small because the gradient of the GPH is very steep at the southern slope of the Himalayas, therefore it is difficult to highlight this belt using only the GPH as criterion to discriminate the foothills. Nevertheless, the number of trajectories with the location of strongest updraft on the foothills

(9%) or on the Tibetan Plateau (36%) is slightly enhanced compared to the end points of these trajectories (8%, 34%) (Tab. 3 and Tab. 4).

    The location of $\Delta\Theta_{max}$ depends strongly on the time interval over which the change in potential temperature is calculated. If a very short time interval is chosen, the deduced region of strongest updraft depends only on one input data set at one particular time step. However, the longer the time interval the higher the uncertainty of the location of maximum updraft along the

trajectories. This is because the vertical updraft by convection in CLaMS trajectories is driven by vertical velocities deduced from ERA-Interim and is not restricted to very narrow regions (e.g. see Fig. 4). Therefore, the locations of $\Delta\Theta_{max}$ are spread out increasingly over the region of the southern slope of the Himalayas with an increasing time interval. To avoid the uncertainty caused by a larger time interval and because of the fact that convection occurs on a time frame of a few hours, we decided to use a time interval of 18 hours. This choice also ensures that at least 3 input data sets from ERA-Interim (with 6 hour resolution)

are included to localise the preferred regions of strongest updraft in the region of the Asian monsoon.

## 4.2   Case 2: No ATAL on 15 August

On 15 August 2016 no ATAL was detected by the COBALD measurements. To compare this flight with the other cases where an ATAL was probed during August 2016, CLaMS backward trajectories are initialised in a pressure range between $140\,\mathrm{hPa}$ to $92\,\mathrm{hPa}$ ($365\,\mathrm{K}$ to $389\,\mathrm{K}$). In this pressure range 45% of the trajectories are from the BL, that is 20% less than the BL contribution

of Case 1 (64%). Therefore, much higher contributions of the UT (30%) and LT (20%) as well as from the LS (5%) are found on 15 August in contrast to Case 1 (22%, 14%, 0.3%) (see Tab. 2). The lesser contribution of the BL can be explained by the location of the anticyclone related to the location of the measurements. On 6 August 2016 the Asian monsoon anticyclone was over Nainital, while on 15 August Nainital was located between the western and the eastern mode of the anticyclone (see Fig. 3).

The 40-day backward trajectories for the 15 August 2016 are shown in Fig. 7. In contrast to Case 1 no pronounced individual convection events are noticeable in the trajectories from BL and LT on 15 August. Here, an alternating up and downward transport is found along most of the trajectories below $360\,\mathrm{K}$ and only a few trajectories show a strong upward transport (convection) up to $360\,\mathrm{K}$ within $1-2$ days.





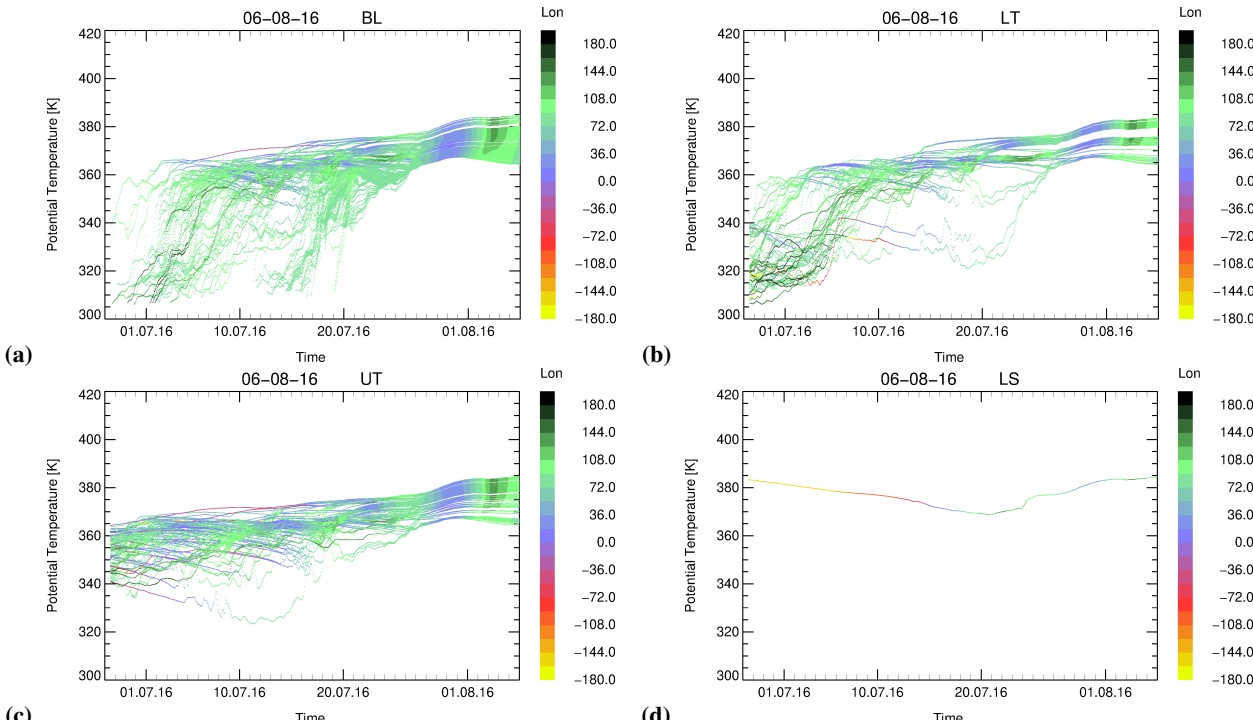

**Figure 4.** 40-day backward trajectories initialised on 6 August 2016 at the location of the ATAL measurement (see Tab. 1). The trajectories are sorted by air mass origin in the boundary layer (BL), in the lower troposphere (LT), in the upper troposphere (UT) and in the lower stratosphere (LS). The fractions of the different atmospheric height layers (BL, LT, UT, LS) to the composition of the ATAL are shown in Tab. 2. The colour indicates the location of the trajectory in longitude.

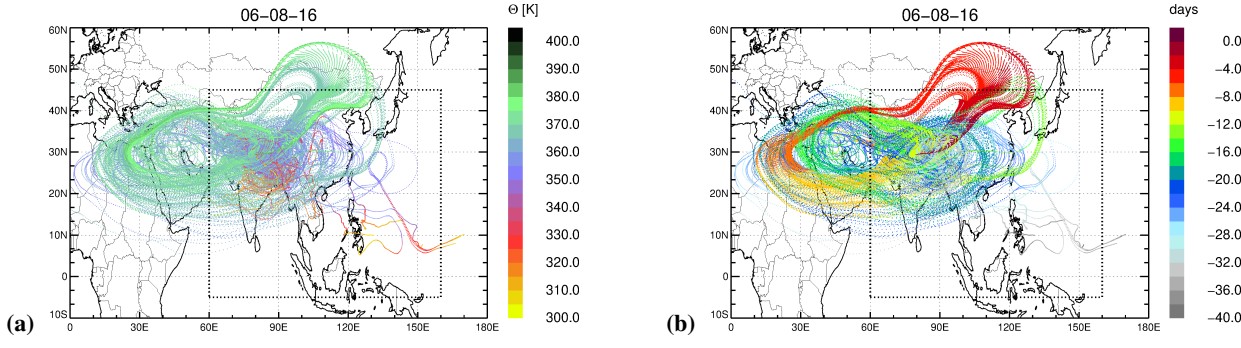

**Figure 5.** 40-day backward trajectories initialised within the ATAL on 6 August 2016 originating in the BL (same trajectories as shown in Fig. 4a) colour-coded by potential temperature **(a)** and by days back from 6 August 2016 **(b)**. In a latitude-longitude box from 60°E to 160°E and from 5°S to 45°N indicated by dashed lines the air mass origin in the BL is further analysed (see Fig. 6, Tab. 3).





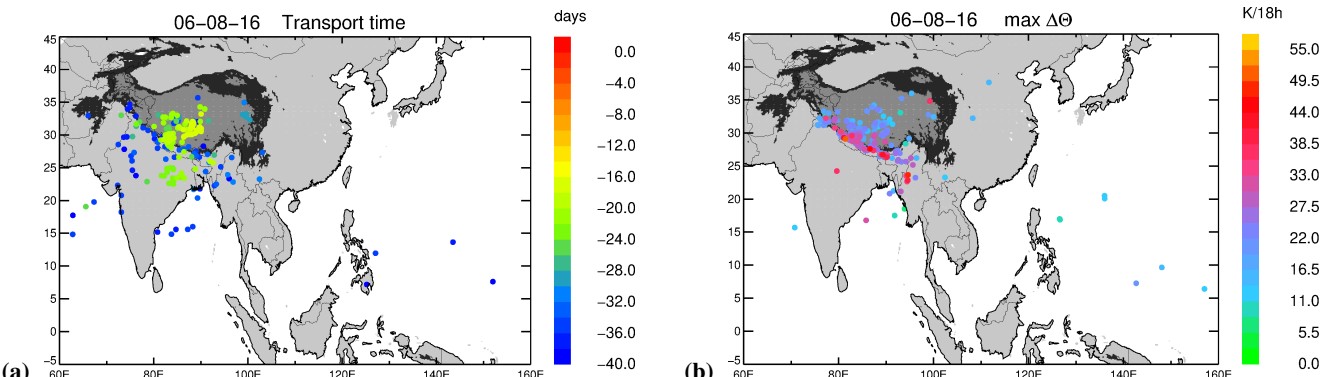

**Figure 6. (a)** Location of end points of the trajectories, initialised on 6 August within the ATAL, where they reach the model boundary layer (BL) within 40 days. The end points are colour-coded by the transport time between BL and measurement. **(b)** Location of the strongest updraft along the 40-day backward trajectories colour-coded by the strongest change in potential temperature within 18 hours [in K 18h$^{-1}$]. The Tibetan Plateau (dark grey), foothills (black), the remaining continental area (Land; light grey) and maritime area (Ocean; white) are indicated (see Tab. 3).

Depending on the altitude of the measurements two branches of trajectories are found (Fig. 7a, b and c); the higher ones are from eastern and the lower ones from western longitudes. Focusing on the trajectories from the BL (Fig. 8), in the western branch of the trajectories the air masses are transported around the western mode of Asian monsoon anticyclone, while air masses from the eastern branch are coming from the Pacific ocean and are transported along the south-

5 ern edge of the eastern mode of the anticyclone to the measurement location in Nainital. During the first half of August several tropical storms occurred in the western Pacific. Typhoon Omais was active between 2−12 August 2016 (https://www.jma.go.jp/jma/jma-eng/jma-center/rsmc-hp-pub-eg/besttrack.html; storm ID: 1605 (last access: 26 May 2020) and https://en.wikipedia.org/wiki/2016_Pacific_typhoon_season (last access: 26 May 2020)) and impacted the sounding on 15 August (see Fig. 9).

10 Fig. 9a shows the location of the end points of back-trajectories in the BL colour-coded by the transport time from the BL to the location of the measurement. Lesser contributions from the Tibetan Plateau (17%) and the foothills (3%) are found compared to Case 1 (34%, 8%). Due to the dynamics of the bi-modal anticyclone the locations of the end points in the BL are much more widely spread over Asia (e.g. Pakistan, Afghanistan, China), the western Pacific, and the residual Earth's surface (≈5 %; see Tab. 3) as compared to Case 1. In Case 1, the end points are more clustered demonstrating the more frequent

15 occurrence of individual convection events. A few trajectories ending on the Tibetan Plateau have short transport times of about 10 to 12 days; these are shorter times than found for Case 1 (Fig. 6) and are caused by convection between 2 and 5 August.





| No. | Date | BL % | Tibet % | Foothills % | Land % | Ocean % | Residual % |
|---|---|---|---|---|---|---|---|
| NT001 | 02-08-16 | 47.0 | 18.5 | 3.0 | 16.9 | 6.9 | 1.8 |
| NT002 | 03-08-16 | 39.1 | 13.8 | 2.4 | 8.4 | 9.1 | 5.5 |
| NT003 | 05-08-16 | 22.3 | 7.5 | 1.7 | 8.0 | 4.1 | 1.0 |
| NT004 | **06-08-16** | 64.2 | 33.5 | 7.5 | 17.7 | 4.2 | 1.3 |
| NT005 | 08-08-16 | 54.3 | 22.6 | 3.8 | 21.9 | 4.6 | 1.3 |
| NT007 | 11-08-16 | 40.9 | 22.3 | 2.6 | 14.1 | 1.5 | 0.4 |
| NT009 | 12-08-16 | 28.7 | 14.3 | 1.1 | 7.6 | 4.5 | 1.3 |
| NT011 | **15-08-16** | 44.8 | 16.9 | 2.7 | 16.2 | 5.2 | 3.8 |
| NT015 | 17-08-16 | 31.8 | 16.4 | 2.6 | 7.7 | 3.5 | 1.5 |
| NT017 | **18-08-16** | 45.1 | 3.7 | 0.6 | 7.2 | 29.5 | 4.1 |
| NT018 | 19-08-16 | 27.1 | 7.7 | 1.1 | 8.8 | 6.7 | 2.8 |
| NT023 | 21-08-16 | 29.6 | 3.2 | 1.2 | 3.2 | 13.2 | 8.8 |
| NT025 | 23-08-16 | 63.7 | 23.6 | 3.0 | 19.9 | 14.5 | 2.7 |
| NT027 | 26-08-16 | 13.3 | 2.5 | 0.0 | 3.3 | 6.7 | 0.8 |
| NT029 | 30-08-16 | 29.6 | 6.1 | 2.3 | 7.9 | 11.5 | 1.8 |
| NT033 | 10-11-16 | 46.1 | 0.0 | 0.0 | 4.1 | 25.8 | 16.3 |
| NT034 | 11-11-16 | 13.9 | 0.0 | 0.0 | 0.0 | 1.0 | 12.9 |

**Table 3.** Fraction of 40-day backward trajectories (for all soundings in August 2016), which are ending in the boundary layer (BL) separated into the Tibetan Plateau (Tibet), foothills (Foothills), the remaining continental area (Land) and the maritime area (Ocean) in a latitude-longitude box from 60°E to 160°E and from 5°S to 45°N shown in Fig. 6, and finally all remaining trajectories reaching the BL outside the latitude-longitude box (Residual). The fraction 100% - BL% are those trajectories, which are from the LT, UT and LS (i.e. do not reach the BL within 40 days as shown in Tab. 2). The Tibetan Plateau is defined as the altitude above a geopotential $\Phi$ of $40000\,\mathrm{m^2\,s^{-2}}$ (about 4000 m), the foothills as $21000 < \Phi < 40000\,\mathrm{m^2\,s^{-2}}$ (about $2000-4000\,\mathrm{m}$), the remaining continental area as $\Phi < 21000\,\mathrm{m^2\,s^{-2}}$ (below about 2000 m).

Similar as for Case 1, in Case 2 the location of the end points in the model BL (Fig. 9a) and of the strongest updraft (Fig. 9b) differ substantially. A cluster of locations with an updraft larger than $25\,\mathrm{K\,18h^{-1}}$ is found at the southern edge of the Himalayas, over Myanmar, and west of the Tibetan Plateau.

In Fig. 9, the location of the end points in the model BL of the strongest updraft for trajectories started in the cirrus cloud found between 376 and 381 K potential temperature on 15 August (see Fig. 2) are marked by a star symbol. End points in the BL of the cirrus cloud are found in different regions, demonstrating that the air masses in the cirrus cloud are a mixture of different origins. Thus the locations of strongest updraft for trajectories initialised within the cirrus cloud are found over the Pacific as well as over continental areas. Cirrus formation as well as likely the details of transport pathways that lead to cirrus formation are not represented in our trajectory calculations based on ERA-Interim reanalysis.





| No. | Date | BL % | Tibet % | Foothills % | Land % | Ocean % | Residual % |
|---|---|---|---|---|---|---|---|
| NT001 | 02-08-16 | 47.0 | 17.2 | 9.0 | 17.8 | 2.8 | 0.3 |
| NT002 | 03-08-16 | 39.1 | 15.3 | 4.3 | 9.8 | 9.3 | 0.5 |
| NT003 | 05-08-16 | 22.3 | 9.0 | 3.4 | 8.0 | 1.7 | 0.2 |
| NT004 | **06-08-16** | 64.2 | 36.1 | 8.8 | 15.8 | 3.4 | 0.0 |
| NT005 | 08-08-16 | 54.3 | 25.9 | 9.3 | 17.2 | 1.8 | 0.1 |
| NT007 | 11-08-16 | 40.9 | 24.9 | 3.7 | 11.5 | 0.7 | 0.0 |
| NT009 | 12-08-16 | 28.7 | 11.4 | 7.1 | 6.9 | 3.0 | 0.2 |
| NT011 | **15-08-16** | 44.8 | 16.0 | 5.2 | 17.8 | 4.3 | 1.6 |
| NT015 | 17-08-16 | 31.8 | 15.7 | 5.5 | 6.6 | 3.7 | 0.3 |
| NT017 | **18-08-16** | 45.1 | 3.4 | 1.6 | 6.4 | 27.2 | 6.5 |
| NT018 | 19-08-16 | 27.1 | 7.6 | 3.7 | 8.8 | 5.4 | 1.6 |
| NT023 | 21-08-16 | 29.6 | 3.2 | 0.8 | 3.2 | 17.6 | 4.8 |
| NT025 | 23-08-16 | 63.7 | 24.8 | 7.9 | 22.4 | 7.3 | 1.5 |
| NT027 | 26-08-16 | 13.3 | 3.3 | 0.0 | 1.7 | 8.3 | 0.0 |
| NT029 | 30-08-16 | 29.6 | 6.9 | 3.1 | 7.1 | 12.5 | 0.0 |
| NT033 | 10-11-16 | 46.1 | 0.0 | 0.0 | 2.6 | 31.2 | 12.3 |
| NT034 | 11-11-16 | 13.9 | 0.0 | 0.0 | 0.2 | 0.8 | 12.9 |

**Table 4.** Same as Tab. 3, but for the location of strongest updraft along the 40-day backward trajectories ending in the BL. The strongest updraft is calculated by the maximum change in potential temperature within 18 hours along the trajectories. The location of the maximum updraft within the 18 hours is then calculated as the mean location of the trajectory within 18 hours.

### 4.3 Case 3: Typhoon influence on 18 August

On 18 August the ATAL occurs over a broader potential temperature range from 362 K to 422 K compared to 6 August (364 K to 388 K, Case 1), however the the ATAL intensity ($\overline{BSR}_{455}$=1.065) is lower compared to Case 1 ($\overline{BSR}_{455}$=1.083) (Fig. 2 and Tab. 1). The contributions from the BL are 45% and similar to Case 2, but much lower than for Case 1 (64%, Tab. 2). For Case 3, a stratospheric contribution of 15% is found, which is much higher than for Case 1 (0.3%) and Case 2 (5%) because of the considered top level of potential temperature (Case 1 up to 388 K, Case 2 up to 389 K and Case 3 up to 422 K) (see Figs. 4d, 7d and 10d). The stratospheric contributions are air masses from the northern extra-tropical lower stratosphere. These air masses are transported along the subtropical jet and subsequently were slowly uplifted in the anticyclonic large-scale upward spiral around the Asian monsoon anticyclone by diabatic heating. In general, the higher within the upward spiralling range at the top of the Asian monsoon anticyclone, the more contributions from the stratospheric background are found (Vogel et al., 2019).

Fig. 10a shows very strong convection between 6 and 15 August with strong updraft within a few days up to 360 K. These air masses originate from the western Pacific and Bay of Bengal and are transported from the western Pacific directly to Nainital





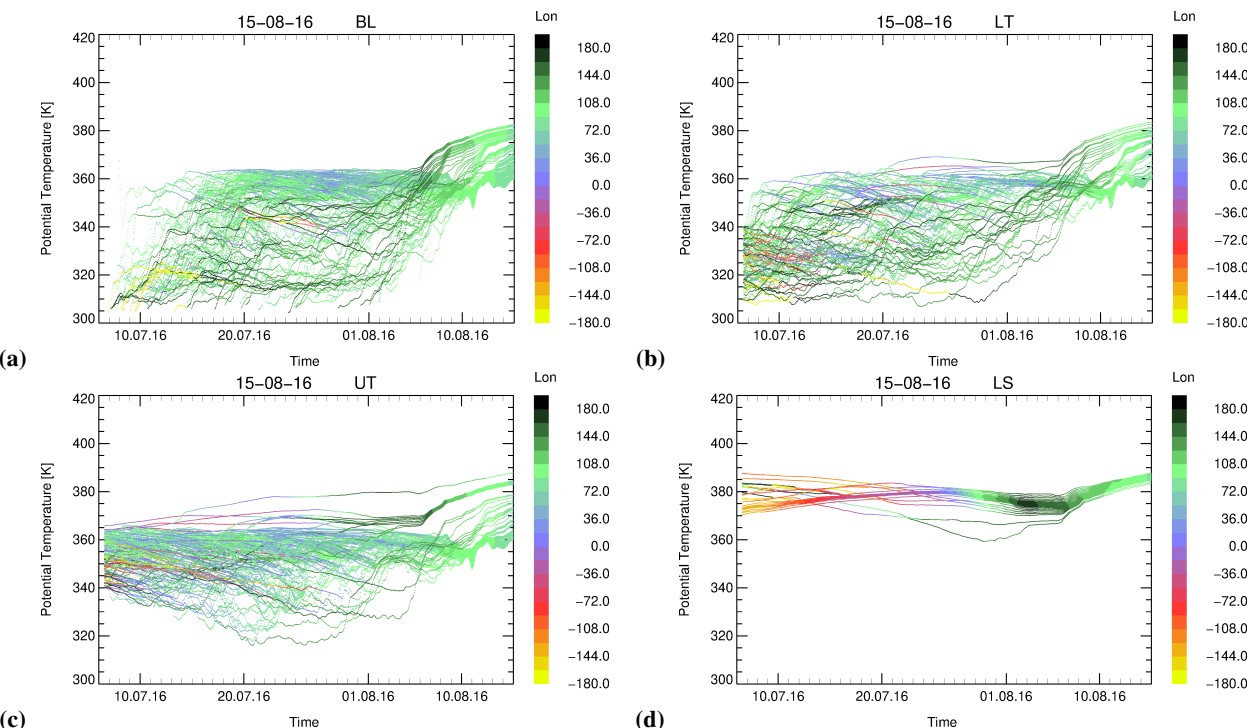

**Figure 7.** The same as Fig. 4, but for Case 2 the no ATAL case on 15 August 2016. The trajectories are calculated in a pressure range between 140 hPa to 92 hPa (365 K to 389 K).

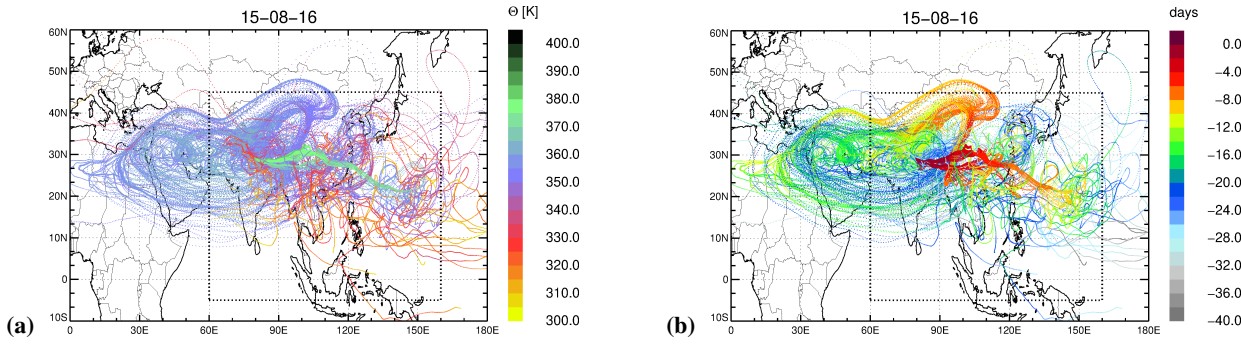

**Figure 8.** The same as Fig. 5, but for the balloon flight on 15 August 2016 (Case 2). In addition to Case 1 shown in Fig. 5, the impact of tropical cyclone activity in the western Pacific is found in Case 2.

(Fig. 11). Another branch of BL air from the western Pacific is transported around the outer edge of the Asian monsoon anticyclone to Nainital (Fig. 11).





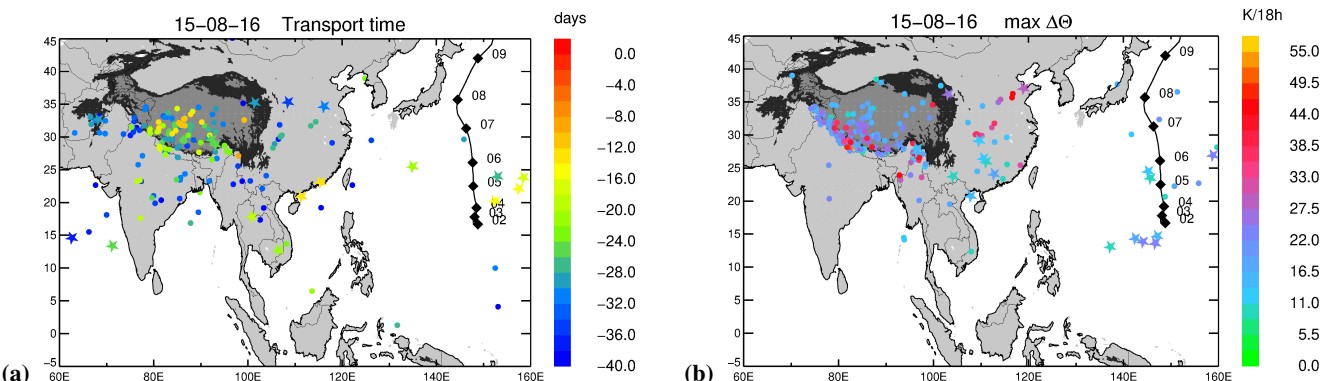

**Figure 9.** The same as Fig. 6, but for the no ATAL case on 15 August 2016. Locations of end points and locations of the strongest updraft of trajectories initialised within the cirrus cloud are indicated by a star symbol. The 40-day backward trajectories are impacted by typhoon Omais. Its storm track is shown (black line) and the location at noon time is additionally marked (black diamonds); the numbers indicate the day in August 2019.

The location of the end points in the BL for Case 3 are shown in more detail in Fig. 12. The major fraction of the BL contribution is from the Ocean (30%) in contrast to Case 1 (4%) and Case 2 (5%). Also the location of $\Delta\Theta_{max}$ is found in the western Pacific, however at slightly different locations as the end points. The strong updraft over the Pacific is caused by tropical cyclone activity. The typhoon Nida was active between 29 July and 3 August over the western Pacific (https://www.jma.go.jp/
5    jma/jma-eng/jma-center/rsmc-hp-pub-eg/besttrack.html; storm ID: 1604 (last access: 26 May 2020) and https://en.wikipedia.org/wiki/2016_Pacific_typhoon_season (last access: 26 May 2020)) and impacted the balloon sounding on 18 August 2016. Thus, in Case 3 polluted air masses within the ATAL layer measured below 400 K are diluted by air from the maritime boundary layer. At higher potential temperature levels (above 400 K) mixing with air masses from the stratosphere occurs.

In Fig. 12, the location of the end points in the model BL of the strongest updraft for trajectories started in the cirrus cloud
10   found between 370 and 373 K potential temperature on 18 August (see Fig. 2) are marked by a star symbol. End points in the BL of the cirrus cloud are found in different regions, similar as for the cirrus on 15 August, demonstrating that the air masses in the cirrus are a mixture of different origins as well as in the ATAL itself. As mentioned in Sect. 4.2, cirrus formation as well as likely the details of transport pathways that lead to cirrus formation are not represented in our CLaMS trajectory calculations.



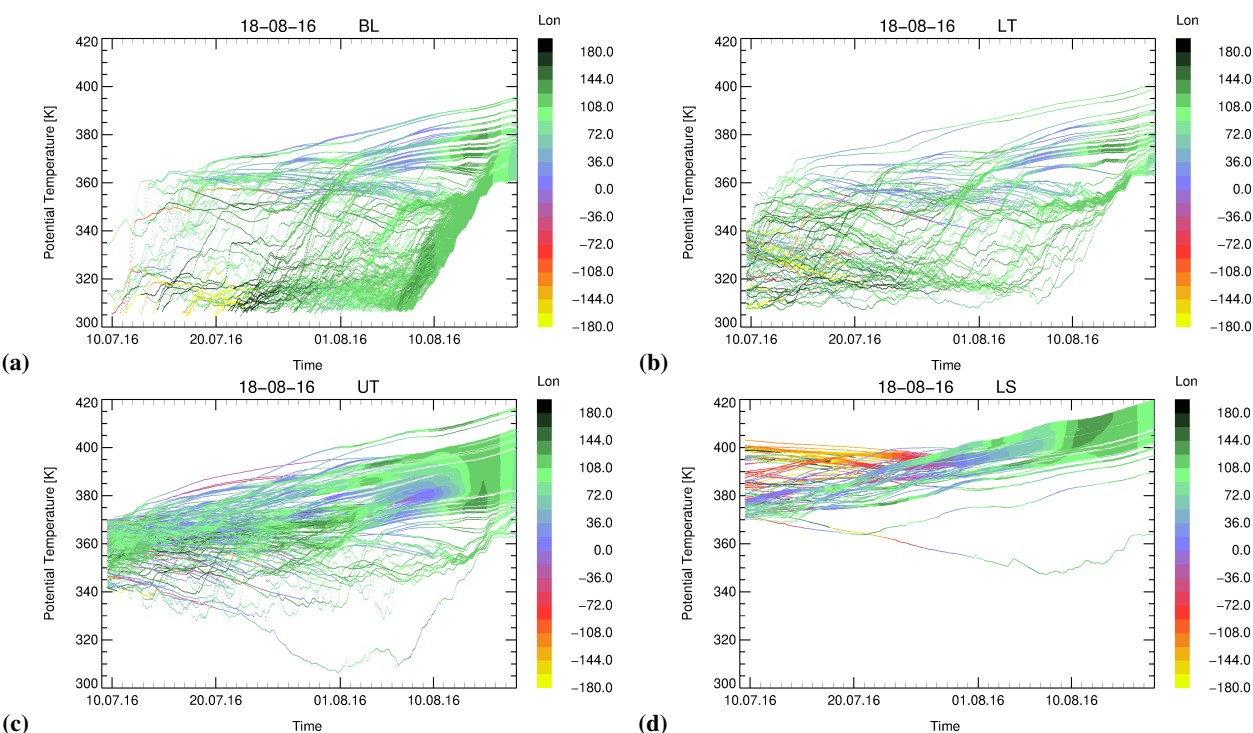

**Figure 10.** The same as Fig. 4 and Fig. 7, but for Case 3 the typhoon-influenced ATAL on 18 August 2016. The trajectories are calculated in a potential temperature range between 150 hPa to 75 hPa (362 K to 422 K).

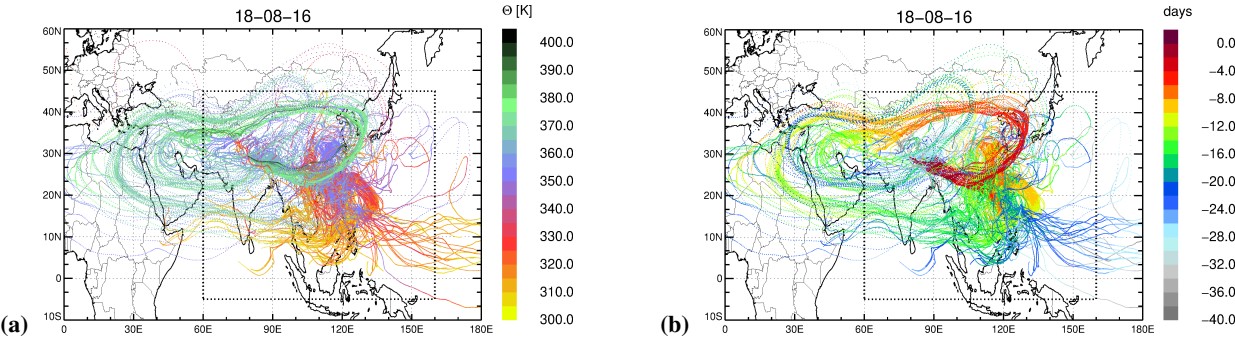

**Figure 11.** The same as Fig. 5 and Fig. 8, but for the balloon flight on 18 August 2016 heavily influenced by a typhoon in the western Pacific (Case 3).

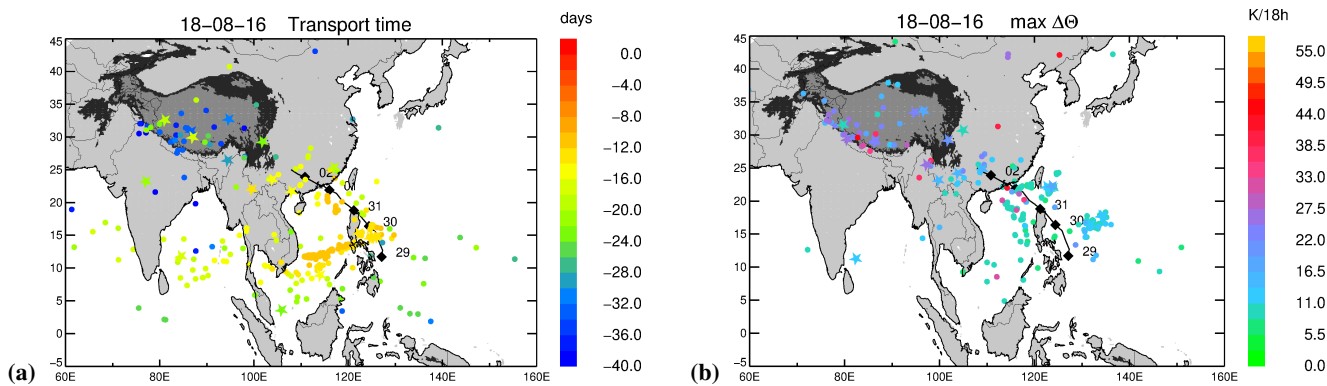

**Figure 12.** The same as Fig. 6 and Fig. 9, but for the typhoon-influenced case on 18 August 2016 (Case 3). Locations of end points and locations of the strongest updraft of trajectories initialised within the cirrus cloud which was found within the ATAL are indicated by a star symbol. The 40-day backward trajectories are impacted by typhoon Nida. Its storm track is shown (black line) and the location at noon time is additionally marked (black diamonds); the numbers indicate the day in July and August 2019.





## 4.4 Results for all flights

In Sects. 4.1, 4.2, and 4.3, the fractions of the different atmospheric height layers (BL, LT, UT, LS) contributing to the composition of air masses within the ATAL were discussed in detail for three specific days, namely the 6, 15 and 18 August 2016. In Fig. 13 (top), the fractions of the different atmospheric height layers for all flights in August and November 2016 are shown as a bar chart (see Tab. 1 and Tab. 2) for 40-day backward trajectories. Due to the strong variability of the vertical extent of the ATAL and the variability of cirrus clouds marking the bottom of the ATAL for certain days, the number of back-trajectories varies strongly from a number of 119 to up to 704. For better comparison the fractions of the different atmospheric height layers are normalised by the total number of trajectories for each day (Fig. 13, top). The fractions from the BL are between 14% and 64% and from the LS between 0% and 33%. Thus there is a strong variability of the fractions of the different atmospheric height layers contributing to the ATAL as well as for the no ATAL cases in August and post-monsoon cases in November within the corresponding pressure levels between 140 and 92 hPa. However, from the analysis of different atmospheric height layers no clear relation between these height layers and the occurrence of ATAL was found, that may explain the difference between the ATAL and no ATAL cases as well as between ATAL and post-monsoon soundings in November.

During the monsoon season air masses from the BL accumulate within the Asian monsoon anticyclone, thus in the altitude range of the ATAL. Therefore, back-trajectory calculations with a length of 60 and 80 days were also performed to consider the sensitivity of our results regarding the trajectory length. In general the fractions from the BL contributing to the ATAL increase with trajectory time. The fractions from the BL range between 14% to 64% for 40 days (Fig. 13, top), from 40% to 83% for 60 days and from 56% to 90% for 80 days. There is an increase of the fractions from the BL between 14 and 36 percentage points between 40 and 60 days (Fig. 13, middle) and between 6 and 25 percentage points between 60 and 80 days (Fig. 13, bottom). Simultaneously the fractions from the UT decrease with time. They range between 17% to 58% for 40 days, from 8% to 28% for 60 days and from 6% to 21% for 80 days (see Fig. 13). A decrease is found of the fractions from the UT between 5 and 38 percentage points between 40 and 60 days (Fig. 13, middle) and between 0 and 12 percentage points between 60 and 80 days (Fig. 13, bottom). The fractions from the LT do not change significantly with increasing trajectory length. They vary between −7 and 8 percentage points between 40 and 60 days (except for 10 November 2016, here the difference is −21 points) and between −11 and 1 points between 60 and 80 days. The fractions from the LS decrease by up to −11 percentage points between 40 and 60 day and up to −7 points between 60 and 80 days.

To deduce a possible relation between the ATAL intensity and the air mass origin within the boundary layer, Fig. 14a shows a bar chart with the contributions of Tibet, Foothills, Land, Ocean and Residual where the individual measurements are sorted by increasing backscatter ratio $(\overline{BSR}_{455}\text{-}1) \times 100)$ (see Tabs. 1 and 3). The contributions are normalised by the total number of trajectories within the boundary layer for each day for better comparison. The balloon flight on 12 August is excluded because a cirrus cloud was detected between 140 and 92 hPa and therefore no aerosol backscatter ratio could be measured in this pressure range. Because of the low statistics we exclude all flights (11, 21, 23 and 26 August) where the trajectory number is lower than 50% of the maximum number of trajectories (# 704) calculated on 18 August 2016 (these cases are included in Fig. 15).



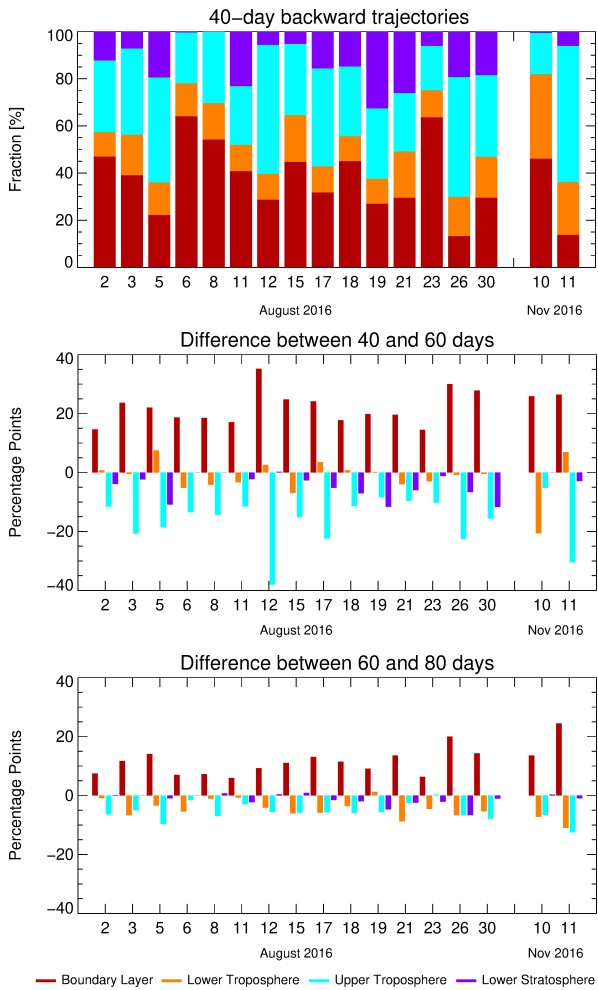

**Figure 13.** The fractions of the different atmospheric height layers (BL, LT, UT, LS) for all flights in August and November 2016 normalised by the total number of trajectories of each day calculated from 40-day backward trajectories (top) (see Tab. 2), the difference between 40 and 60 days (middle) backward trajectory calculations as well as between 60 and 80 days (bottom).

There is a lot of variability of the air mass origin within the boundary layer contributing to ATAL cases characterised by a backscatter ratio $\overline{BSR}_{455}$ larger than 1.058 (Fig. 14, top). Strong ATAL cases with $\overline{BSR}_{455}$ values larger than 1.067 show fractions of air mass origin in the boundary layer from continental outflow higher than 70% and in particular higher than 30% from Tibet (Fig. 14, top). For weak ATAL cases (18 and 30 August), the $\overline{BSR}_{455}$ values are lower $(1.059-1.065)$ and the fractions from continental outflow is below 70%. For these cases high fractions from maritime boundary layer sources (Ocean $> 12\%-30\%$) are found which are caused by the impact of tropical cyclones on the composition of the ATAL as discussed in Sect. 4.3.





For both post-monsoon cases in November 2016 low $\overline{\mathrm{BSR}}_{455}$ values of 1.034 and 1.037 are measured, these measurements are used as background signal for aerosols in the UTLS (no ATAL has been observed during winter). For these cases the air mass origin is very different, compared to the ATAL cases. In 40-day back-trajectory calculations no contributions from Tibet and Foothills are found for both flights, only a contribution from Land of 4% is found for the balloon flight on 11 November 2016. The origins of air masses in the boundary layer are from Ocean and from Residual mainly from south of 5°S. It is known that during boreal winter efficient transport into the stratosphere is found over the west Pacific and Maritime Continent, caused by strong convection and in addition by the ascending branch of the Walker circulation located over the Maritime Continent (e.g. Bergman et al., 2012; Hosking et al., 2012).

Case 2, the no ATAL case from 15 August 2016, has a $\overline{\mathrm{BSR}}_{455}$ value of 1.023 lower than both November cases, however an air mass origin in the boundary layer is found of around 80% from continental sources (Tibet, Foothills and Land). This BL fraction is similar to fractions from the BL as for the ATAL cases. Case 2 is discussed in detail in Sect. 4.2 and it was shown that on 15 August a bi-modal structure of the anticyclone is found and Nainital is located between the eastern and western part of the anticyclone (Fig. 3). This bi-modal structure is found between 11 and 16 August (not shown here). Thus also the no ATAL case on 12 August and the balloon-flight on 11 August are impacted by this dynamical situation in the UTLS. Therefore, the dynamics in the UTLS seems to be the reason that on 15 August 2016 no ATAL was measured over Nainital, although a similar air mass origin was found on 15 August as for the ATAL cases. Unfortunately, during August 2016 we have only one no ATAL measurement (not completely overlaid by a cirrus cloud), therefore we can not deduce a more general result regarding no ATAL measurements during the peak monsoon season in India in our study. During August 2016, a second period with a bi-modal structure of the anticyclone is found between 27 and 28 August, but here Nainital is located more in the centre of the eastern mode (not shown here).

The change between the ATAL intensity and the air mass origin within the boundary layer (Tibet, Foothills, Land, Ocean and Residual) between 40 and 60 days (Fig. 14b) as well as between 60 and 80 days (Fig. 14c) is also analysed. In general, the fractions of the Residual is increasing with the trajectory length for all balloon flights, except for 11 November. Thus the longer the trajectories, the more older air masses are taken into account to contribute to the ATAL coming from outside the latitude-longitude box from 60°E to 160°E and from 5°S to 45°N. For strong ATAL cases, in general an increase of the fractions from Ocean up to ~7 percentage points are found (Fig. 14b) indicating the impact of maritime convection between 40 and 80 days before the soundings. In contrast, for weak ATAL cases, the fraction of continental convection (Tibet, Foothills, and Land) is increasing, indicating the impact of continental convection between 40 and 80 days before the soundings.

Figs. 6, 9 and 12 show that on 6, 15 and 18 August the strongest updraft locations differ from the endpoint locations of the trajectories. A cluster of strongest updraft locations is found at the southern edge of the Himalayan foothills. Tab. 4 shows the fractions of Tibet, Foothills, Land, Ocean and Residual of the strongest updraft locations within 18 hours for all flights. Our definition of the foothills at the southern edge of the Himalaya results in this region being only a small belt identified by the orography. Thus not all trajectories having their strongest updraft at the southern edge of of the Himalayan foothills are captured by our definition of the foothills and therefore these trajectories are still counted as fractions of Land. Fig. 14d shows the difference between the location of the end points and the location of strongest updraft for 40-day backward calculations.





For some soundings (3, 5, 6, 8 and 30 August) the fractions from Foothills and Tibet for the location of strongest updraft are greater than for the location of the end points indicating strong convective activity at the Himalayan foothills during early August 2016.

Fig. 15 shows the fractions of Tibet, Foothills, Land and Ocean versus $(\overline{\mathrm{BSR}}_{455}\text{-}1) \times 100$ (see Tab. 1) for the location of the end points of the trajectories (top) (see Tab. 3) and for the location of the strongest updraft (bottom) (see Tab. 4). The fractions are not normalised to the number of trajectories within the BL as was done in Fig. 14. Therefore, in Fig. 15 also the contribution from the other atmospheric layers (LT, UT and LS) to the ATAL are taken into account. Further, the low statistic flights (with a relatively low absolute number of trajectories, 11, 21, 23 and 26 August) are also considered. There is a lot of variability between individual soundings. However, in general an increase of the fractions from Tibet, Foothills and Land is found for increasing ATAL intensity, while the contributions from the Ocean are decreasing for both the location of end points and the locations of strongest updraft. The gradient of the linear fit for the fractions of Tibet and Foothills is somewhat steeper for the locations of strongest updraft compared to the location of the endpoints highlighting the role of the southern Himalayan Foothills and the Tibetan Plateau in the uplift of ATAL aerosols and their chemical precursors from the Earth's surface to ATAL altitudes.

## 5   Discussion

COBALD measurements using the CI, the 940-to-455 nm ratio of the aerosol component of the BSR, can be used to discriminate aerosol and cirrus measurements (see Sect. 2.1). However, in case of a cirrus cloud, the dominant ice particle backscatter does not allow the COBALD sonde to detect, whether aerosol particles coexist with ice particles or not. Krämer et al. (2016) describe two types of cirrus: (1) in-situ origin cirrus observed at the altitudes where they are formed on soluble solution aerosol particles or on solid ice nucleating particles; (2) liquid origin, cirrus which are glaciated liquid clouds lifted from below to the cirrus temperature region. The formation mechanism of the cirrus cloud will have implications for the aerosol concentration in the cirrus layers within the ATAL.

For our back-trajectory calculations thin cirrus clouds within the ATAL are included to infer the origin of air masses contributing to the ATAL. If we assume that these cirrus clouds are of cirrus type (2), the aerosol concentration within the ATAL would not be affected by the formation process of the cirrus particles, however the uplift (convection) of air masses from the lower troposphere could transport enhanced concentrations of aerosol particles to ATAL altitudes (as proposed by Vernier et al., 2018). Vernier et al. (2018) found in some balloon-borne measurements of the ATAL from Hyderabad, India, in summer 2015 that the presence of cirrus is associated with a reduction or a minimum in aerosol concentration possibly caused by aerosol removal processes through the formation of cirrus particles of type (1). Therefore, for both cirrus types, it is important to include the back-trajectory calculations of thin cirrus clouds found within the ATAL to identify the air mass origin of air masses contributing to the ATAL. Further, depending on the lifetime of the cirrus clouds within the ATAL, the occurrence of cirrus particles (e.g. through sedimentation or uptake on the ice surface) could have implication on the ATAL even when the cirrus particles are no longer present.





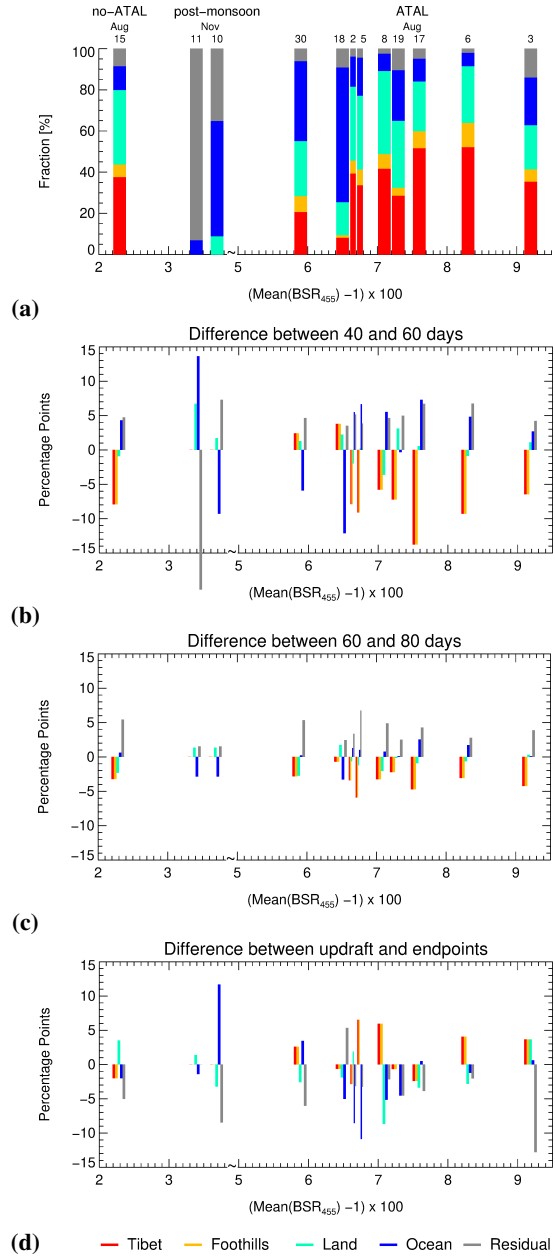

**Figure 14.** Contributions of Tibet, Foothills, Land, Ocean and Residual to the boundary layer (BL) sorted by increasing backscatter ratio $(\overline{BSR}_{455}\text{-}1) \times 100$) (see Tabs. 1 and 3) for 40-day backward trajectories **(a)**. The contributions are normalised by the total number of trajectories within the boundary layer for each day. The days of the soundings in August and November are indicated above the top of the bars. For better representation, the range between 4.0 and 5.0 is removed from the x-axis. Low statistics flights (11, 21, 23 and 26 August) where the trajectory number is lower than 50% of the maximum number of trajectories (# 704) calculated on 18 August 2016 are not shown as well as the 12 August, when the UTLS is filled by a 5 km thick cirrus cloud. The difference between 40 and 60-day **(b)** and between 60 and 80-day **(c)** backward trajectory calculations as well the difference between the location of the end points and the location of strongest updraft **(d)** for 40-day backward calculations is also shown.



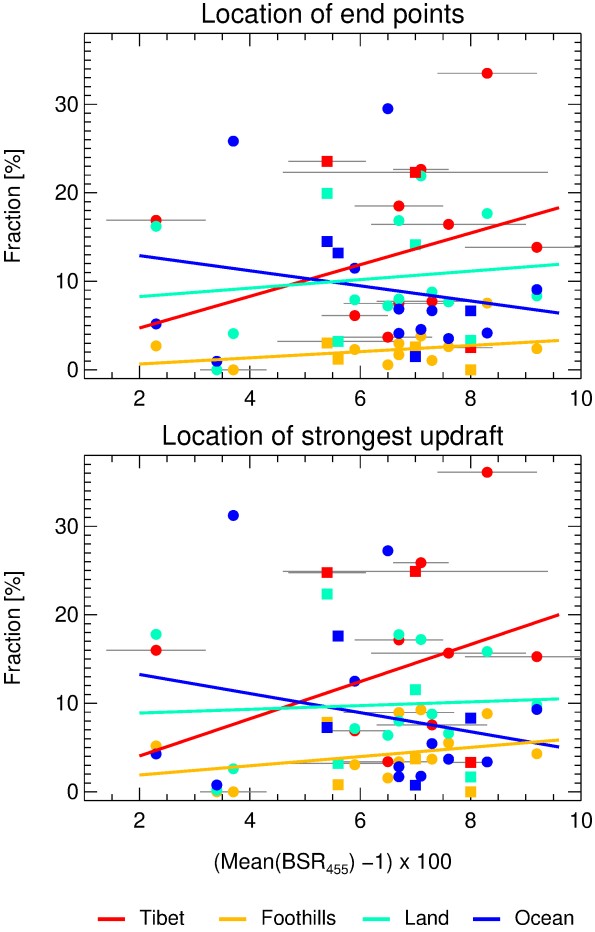

**Figure 15.** Fractions of Tibet, Foothills, Land and Ocean versus ($\overline{\mathrm{BSR}}_{455}$-1) × 100 (see Tab. 1) for the location of the end points of the trajectories (top) (see Tab. 3) and for the location of the strongest updraft (bottom) (see Tab. 4). The standard deviation ($\sigma$) of the backscatter ratio is indicated for each sounding (grey lines). Low statistics flights (11, 21, 23 and 26 August) where the trajectory number is lower than 50% of the maximum number of trajectories (# 704) calculated on 18 August 2016 are also shown (indicated by squares instead of circles). For each region a linear fit was calculated using all measurements except the one on 12 August, when the UTLS is filled by a 5 km thick cirrus cloud (Tab. 1). An increase of the fractions from Tibet, Foothills and Land is found for increasing ATAL intensity, while the contributions from the Ocean decrease for both the locations of end points and the locations of strongest updraft.



We would like to emphasise that the calculation of $\overline{BSR}_{455}$ (see Tab. 1) is based on binned data (with an altitude resolution of $\sim$25 m in the UTLS). Small cirrus clouds found within the ATAL are excluded for the calculation of the mean value of $\overline{BSR}_{455}$, because here the $BSR_{455}$ value for cirrus is much higher than for aerosol (Sect. 3.1). However, the calculation of the fraction of air masses uses the high-resolution measurement data (with a resolution of 1 sec) for back-trajectories including the altitude

ranges where small cirrus clouds within the ATAL were detected. In the altitude range of small cirrus clouds, the aerosol concentration could be enhanced by convection or reduced by aerosol removal processes such as in-situ cirrus formation. In both cases it is important to include the air masses in the altitude range where thin cirrus clouds were detected to identify the origin of air masses contributing to the ATAL.

However, cirrus clouds detected directly below the ATAL are excluded in our trajectory analysis, even if aerosol and cirrus

coexisted here the bottom of the ATAL is highly uncertain. Therefore, in our analysis there is an uncertainty in the vertical altitude range of the ATAL. To solve this uncertainty balloon-borne measurements in addition to COBALD would be required such as the measurements of the particle size distribution (e.g. using an optical particle spectrometer, POPS; Gao et al. (2016)).

## 6   Conclusions

We presented a series of balloon-borne measurements performed by the COBALD instrument conducted in Nainital, Northern

India, during August and November 2016. The $BSR_{455}$ measurements show a strong day-to-day variability of the altitude, the vertical extent and the backscatter intensity of the ATAL at UTLS altitudes over Nainital in August 2016. This variability is not visible in the climatological mean values of the ATAL derived by satellite observations. Further, there are frequent observations of cirrus layers embedded in the ATAL; depending on the duration and formation mechanism of the cirrus layers, there will an impact of cirrus on the properties of the aerosol particles constituting the ATAL. In general the $BSR_{455}$ values measured in

August are higher than values measured during post-monsoon in November, which represent the aerosol background during winter. However, there is one observation on 15 August showing no ATAL (here the $BSR_{455}$ is lower than in November).

Lagrangian back-trajectory calculations were performed using the CLaMS model driven by ERA-Interim reanalysis to identify the air mass origin in the model boundary layer and the transport pathways of air parcels contributing to the ATAL over Nainital in August 2016. There is a variety of factors impacting the variability of the ATAL: continental convection, tropical

cyclones (maritime convection), dynamics of the anticyclone and stratospheric intrusions. All these factors contribute to the observed day-to-day variability of the ATAL found over Nainital in August 2016.

We found that air masses contributing to the ATAL are a mixture of air masses from different origins and not exclusively from the BL. The trajectories originating from the BL are separated according to geopotential height and further, for low geopotential heights, between continental and maritime sources. Strong ATAL cases have high contributions from the Tibetan

Plateau (Tibet), the remaining continental (Land) area and the foothills of the Himalayas (Foothills). Weaker ATAL cases have higher contributions from the maritime area (Ocean) caused by the impact of tropical cyclones. Here, mixing with unpolluted air masses from the maritime boundary layer yield a dilution of the ATAL. The no ATAL case of the 15 August 2016 has similar contributions from the Tibetan Plateau (Tib), the remaining continental (Land) area and the foothills of the Himalayas





(Foothills) as the strong ATAL cases, however this day there is a bi-modal structure of the anticyclone and Nainital is located between the western and eastern mode of Asian monsoon anticyclone. We hypothesise that the main driver for no ATAL cases are the dynamics of the Asian monsoon anticyclone. However, we have only one no ATAL measurement in August 2016 (partly influenced by cirrus), therefore more balloon-borne measurements are necessary to validate this hypothesis.

Moreover, we calculated the locations of strongest updraft along the backward trajectories within 18 hours and found a cluster of such locations at the southern edge of the Himalayan foothills over northern India, Nepal and Bhutan. Further for balloon-measurements impacted by tropical cyclones locations of strongest updraft are also found over the western Pacific ocean. Within the ATAL, a mixture of air parcels with different transport times is found. The transport times from the Earth's surface to ATAL altitudes vary strongly. The shortest transport times found for 6, 15, and 18 August are between 10 and

15 days. These trajectories are originating either on the Tibetan Plateau or at the location of tropical cyclones in the western Pacific. Shorter transport times below 10 days are only found for the 12 (when the UTLS is filled by a 5 km thick cirrus cloud) and 17 August for air masses originating on the Tibetan Plateau.

     Finally, CLaMS backward trajectory calculations identify the transport pathways from the Earth's surface to ATAL altitudes. Very fast uplift in a convective range transports air masses up to the top of the convective outflow level ($\sim$360 K) within a few

days. Subsequently, the air parcels are slowly uplifted by diabatic heating within a large-scale upward spiral driven by the anticyclonic flow in the UTLS over the Asian monsoon region from about 360 K up to ATAL altitudes. Over Nainital in summer 2016, a maximum ATAL altitude of 422 K (75 hPa) was measured. This slow uplift caused by diabatic heating in a large-scale upward spiral is consistent with concepts referred to as 'an upward spiralling range' (Vogel et al., 2016) or as 'a confined lower stratosphere' (Brunamonti et al., 2018).

Our study contributes to deducing the source regions of emissions of precursors of ATAL particles at the Earth's surface and their transport pathways to the UTLS which is important to develop recommendations for regulations of anthropogenic surface emissions of ATAL precursors. In a recent study, Fadnavis et al. (2019b) argue that further increasing industrial emissions in Asia will lead to a wider and thicker ATAL having the potential to amplify the severity of droughts in India. Severe droughts would have fatal consequences on agriculture on the Indian subcontinent and therefore would result in strong socio-economic

impacts in one of the most densely populated parts of the world.

*Code and data availability.* The CLaMS code is available at the GitLab server: https://jugit.fz-juelich.de/clams/CLaMS (last access: 20 June 2019) as well as at the on-line repository Zenodo via https://doi.org/10.5281/zenodo.2632683. The typhoon tracks can be obtained at https://www.jma.go.jp/jma/jma-eng/jma-center/rsmc-hp-pub-eg/besttrack.html. The data of the balloon sounding in Nainital 2016 used in this paper are available upon request to Sreeharsha Hanumanthu (s.hanumanthu@fz-juelich.de).





## Appendix A: Colour Index CI



**Figure A1.** COBALD BSR$_{455}$ (blue) and BSR$_{940}$ (red) soundings in the UTLS region versus pressure $60-200$ hPa for each flight in August 2016. In addition, the colour index (CI = (BSR$_{940}$ − 1)/(BSR$_{455}$ − 1); green) and the ice saturation (S$_{ice}$, orange) is shown. Corresponding height calculated using both measured pressure and temperature is also given for orientation (see right-hand y-axis). In our analysis, layers with CI > 7.0 (CI = 7; dotted green line), BSR$_{940}$ ≥ 2 (BSR$_{940}$ = 2; dotted red line) and S$_{ice}$ > 70% (S$_{ice}$ = 0.7; dotted orange line) were rejected as cirrus clouds. The top and bottom boundaries of the ATAL (see Tab. 1) are marked by horizontal dashed black lines, except for 12 and 15 August, as no ATAL was detected on these days (as in Fig. 2).





**Appendix B: The shortwave and longwave channel to detect ATAL**

The aerosol measurements analysed here are based on balloon-borne backscatter measurements employing the COBALD detector, which operates at optical wavelengths of 455 nm (blue visible) and 940 nm (infrared) (Brabec et al., 2012, see main paper). Commonly, the colour index (CI) is defined as the 940-to-455 nm ratio of the aerosol component of the BSR (Cirisan

5 et al., 2014; Brunamonti et al., 2018, see also main paper) and provides some estimate of particle size. The BSR for both wavelength will give an indication of the presence of the ATAL. The same is true for cirrus particles, i.e., they can be detected by both the blue and the red channel, individually. But only by using the CI a clear discrimination of ice (CI > 7) and aerosol (CI < 7) is possible (see main paper), which is the advantage of measuring at two wavelengths.

Here and in previous work (Vernier et al., 2015, 2018; Brunamonti et al., 2018) the ATAL analysis is based on the COBALD

10 455 nm BSR measurements. Indeed, the 455 nm BSR measurement is the preferred channel for the detection of the ATAL. The main reason is that the 455 nm COBALD BSR measurement has a better precision and has a higher signal-to-noise ratio (although BSR is lower at 455 nm than 940 nm, the raw signal is higher at 455 nm). Further, the accuracy is better for 455 nm, as the scale parameter in the data processing is taken directly from the sonde profile. The 940 nm channel builds on that and the BSR values are restricted by the assumption that the CI has to remain in a certain range.

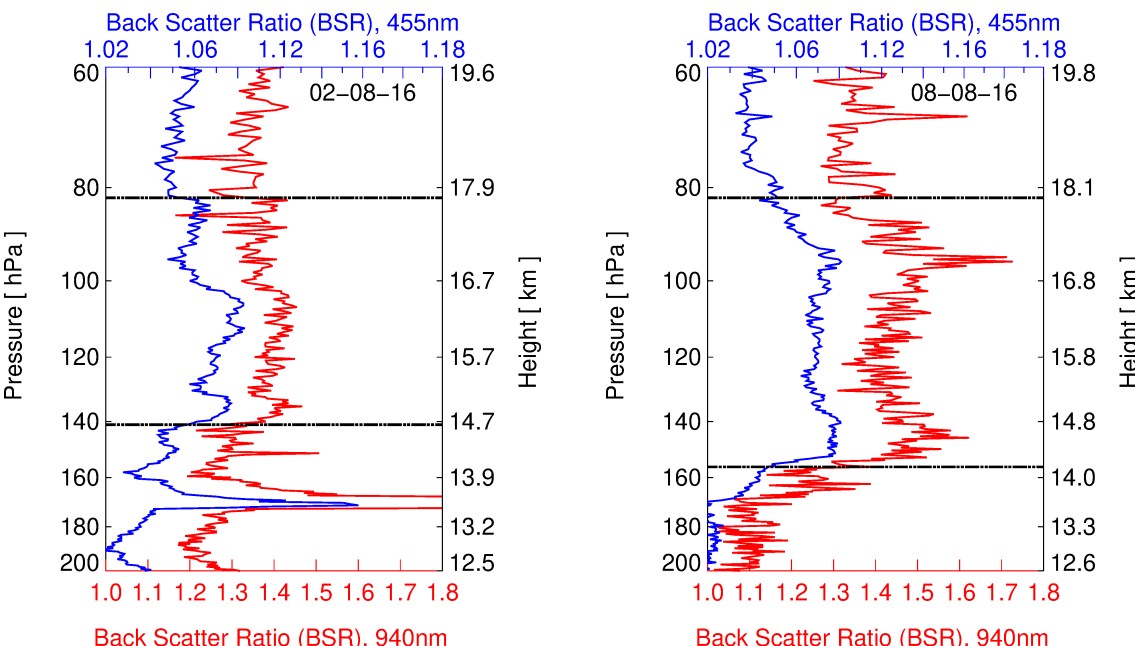

**Figure B1.** COBALD $BSR_{455}$ (blue) and $BSR_{940}$ (red) measurements in the UTLS region versus pressure 60-200 hPa for two flights in August 2016 (2 August, left panel, and 8 August, right panel). Shown are the measured data in pressure bins without further processing (see main paper). The top and bottom boundaries of the ATAL (see Table 1 in the main paper) are marked by horizontal dashed-dot black lines. (Note the different scaling for $BSR_{455}$ and $BSR_{940}$.)



In Fig. B1 we show the measured $BSR_{455}$ (blue line) and $BSR_{940}$ (red line) profiles from the COBALD measurements in Nainital in 2016 for two days in August (with no interference from cirrus layers or spike artefacts). The discussed features are clearly visible; although the ATAL can be detected using both the $BSR_{455}$ and $BSR_{940}$ measurements, the 455 nm channel is less noisy, so it constitutes the preferred channel for the detection of the ATAL in COBALD balloon measurements.

## 5 Appendix C: Removal of 'moon spikes'

The processing of COBALD includes the rejection of 'moon spikes' that may arise due to the oscillatory motion of the payload 60 m below the balloon, when the detector happens to be pointing towards the moon. Moon spikes affect only a tiny fraction of the COBALD data and care is taken not to confuse them with thin cirrus clouds.

To identify those anomalous 'spikes', we used a simple criterion; we consider the signal to be an anomalous spike when
$BSR_{455}$ nm $> 1.12$ and CI $< 7$. These spikes are then removed from the data set, where the condition CI $< 7$ ensures that cirrus clouds will be retained.

Finally, some specific cases with spikes in the $BSR_{455}$ measurements of a very small vertical extent still remained in the binned data, where $BSR_{455} < 1.12$ and $7 < CI < 10$. These specific cases occurred for 3, 15, 21, 23, and 30 August; the corresponding data points were also removed from the data set used here.

*Author contributions.* S.B., T.J. and T.P. coordinated all measurements. The study was conceived by B.V., S.H. and R.M.. The measurements in Nainital were conducted by S.H., S.B., T.J., P.Ö., M.N., B.B.S., K.K., S.S. and T.P.. B.B.S., M.N., S.S. and S.F. provided logistic support for the measurements in Nainital. The trajectory calculations and the associated analysis were conducted by S.H. and B.V., with contributions by D.L.. The results of the study were discussed by all coauthors, with particular contributions by S.B., B.L., T.P., S.H., B.V., and R.M.. The paper was written by B.V., S.H., and R.M., with contributions from all coauthors.

*Competing interests.* The authors declare that no competing interests are present.

*Acknowledgements.* The authors thank Martina Krämer (Research Centre Jülich) for helpful discussions on cirrus formation. The research presented here received funding from the Seventh Framework Programme (FP7/2007–2013) of the European Community under grant agreement no. 603557 as part of the StratoClim project and the Swiss National Science Foundation under project no. 200021-147127. The use of the ECMWF ERA-Interim data is gratefully acknowledged. Support from the Director ARIES and the ISRO ATCTM project is highly
acknowledged regarding the observations at Nainital. S.H. was partly funded by a HITEC (Helmholtz Interdisciplinary Doctoral Training in Energy and Climate Research) fellowship by the Forschungszentrum Jülich and by the German Science Foundation (Deutsche Forschungs-gemeinschaft, DFG) under the DFG project AMOS (HALO-SPP 1294/VO 1276/5-1).





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
