# Peer review of "Strong day-to-day variability of the Asian Tropopause Aerosol Layer (ATAL) in August 2016 at the Himalayan foothills"

_Atmospheric Chemistry and Physics, 2020_

## Referee Comment (RC1) · Anonymous Referee #1 · 16 Jul 2020

Based on COBALD measurements in North India in 2016 August, the variability of the ATAL features is analyzed, and the source regions is simulated with trajectory model - CLaMS. Some interesting results are derived, such as the strong variability of the ATAL's altitude, vertical extend, and aerosol backscatter intensity. Some important transport pathways are identified for different ATAL intensity, such as continental convection and maritime typhoon. The phenomena with no ATAL detected is puzzling. Overall, this manuscript is well written and is recommended to be published in ACP. Minor issues: 1. P2L12: 17km –> 18km 2. P2L19-20: The Asian summer monsoon circulation is affected first by the land-sea contrast, and second by the presence of the Tibetan Plateau. Therefore, this sentence should be modified. 3. P2L21: The

monsoon anticyclone is linked to deep convection in summer over the Indian subcontinent AND OTHER ASIAN MONSOON REGIONS. 4. P3L18-20: The Asian tropopause transition layer in summer is first investigated by Pan et al. (2014). Pan, L. L., L. C. Paulik, S. B. Honomichl, L. A. Munchak, J. Bian, H. B. Selkirk, and H. Vömel, 2014: Identification of the tropical tropopause transition layer using the ozone-water vapor relationship, J. Geophys. Res. Atmos., 119, doi:10.1002/2013JD020558. 5. P4L1-2: How to show the size, radius or diameter? 6. P4L3-4: 0.6W/m2 and 0.5K??? 7. P4L7-10: In some ATAL studies, volcanic eruptions are removed, because volcanic signal is much stronger than ATAL, which will mask the effect of ATAL. 8. P4L28-29: Nitrate aerosol is dominant in the ATAL is first suggested by Gu et al. (2016) by simulation. Gu, Y., H. Liao, and J. Bian, 2016: Summertime nitrate aerosol in the upper troposphere and lower stratosphere over the Tibetan Plateau and the South Asian summer monsoon region, Atmos. Chem. Phys., 16, 6641-6663, doi:10.5194/acp-16-6641-2016. 9. P6L30: CI>7.0, BSR940>2 AND Sice>70% –> OR 10. P6L30: Aerosol layers without cirrus can exist under the condition Sice > 70%. 11. P6L32: Could you provide the vertical range of UTLS? 12. P9L11-14: This "somewhat different picture" can be explained by the results from Pan et al. (2014) as mentioned above. The distribution of lapse-rate minimum levels is compact in potential temperature scale but diffuse in the altitude scale. 13. P10L1: This result can also explained by the results on CPT from Pan et al. (2014). 14. P18L1: The two branches are not easily found in the figure, could you show more clearly? 15. P27L1-6: Why post-monsoon cases are used as background signal for aerosols in the UTLS? Yes, there's no ATAL during winter. But, the general circulation is quite different for summer and winter, so obviously the source regions are different. Could you use cases with no ATAL during summer as background signal? 16. P27: For case 2, beside source regions, other parameters impacting ATAL such as temperature and Sice should also be considered. 17. P28L26: Some in situ measurements show that the aerosols concentration in the middle troposphere is very low, which should be considered in the argument. 18. P31L24-26: Factors impacting the variability of the ATAL include not only source regions, but also other parameters

related the formation and growth of aerosol, the latter should also be considered. 19. P32L1-4: Possibly, satellite data for aerosol can be used. 20. P32L20-22: I think it's too early to talk about the regulations, because we still don't know whether the ATAL existence is good or bad to human beings.

---

## Author Comment (AC1) · 5 Aug 2020

**Author Comment to Referee #1**

**ACP Discussions doi: 10.5194/acp-2020-552-RC1, (Editor - Gabriele Stiller), 'Strong variability of the Asian Tropopause Aerosol Layer (ATAL) in August 2016 at the Himalayan foothills' by Sreeharsha Hanumanthu et al.**
* * *
We thank Referee #1 for the positive review and for important further guidance on how to revise our manuscript. Our reply to the reviewer comments is listed in detail below. Questions and comments of the referee are shown in italics. Passages from the revised version of the manuscript are shown in blue.

*Based on COBALD measurements in North India in 2016 August, the variability of the ATAL features is analyzed, and the source regions is simulated with trajectory model - CLaMS. Some interesting results are derived, such as the strong variability of the ATALs altitude, vertical extend, and aerosol backscatter intensity. Some important transport pathways are identified for different ATAL intensity, such as continental convection and maritime typhoon. The phenomena with no ATAL detected is puzzling. Overall, this manuscript is well written and is recommended to be published in ACP.*

**Minor issues:**

1. *P2L12: 17km → 18km*

   done

2. *P2L19-20: The Asian summer monsoon circulation is affected first by the land-sea contrast, and second by the presence of the Tibetan Plateau. Therefore, this sentence should be modified.*

   We agree and modified the sentence in the revises version of the manuscript as follows:

The dynamics and thermodynamics of the Asian monsoon are caused by the land–sea contrast and are influenced by the orography of the Himalayas and the adjacent mountain ranges (e.g. Turner and Annamalai, 2012, and references therein).

3. *P2L21: The monsoon anticyclone is linked to deep convection in summer over the Indian subcontinent AND OTHER ASIAN MONSOON REGIONS.*

Many thanks for this advice. We revised the sentence as follows in the revised version of the paper:

The Asian monsoon anticyclone is linked to deep convection in summer over south Asia and the associated diabatic heating (Hoskins and Rodwell, 1995; Randel and Park, 2006).

4. *P3L18-20: The Asian tropopause transition layer in summer is first investigated by Pan et al. (2014). Pan, L. L., L. C. Paulik, S. B. Honomichl, L. A. Munchak, J. Bian, H. B. Selkirk, and H. Vömel, 2014: Identification of the tropical tropopause transition layer using the ozone-water vapor relationship, J. Geophys. Res. Atmos., 119, doi:10.1002/2013JD020558.*

We added the following sentences regarding the Asian tropopause transition layer (ATTL) in the revised version of the paper to the introduction:

Pan et al. (2014) identified the tropical tropopause transition layer (TTL) using chemical tracer-tracer relationships in the tropics and over the Asian monsoon. Their comparison shows that the tracer-identified transition layer over the Asian monsoon is similar to the TTL, although the ATTL is located at higher potential temperature levels. However, during the monsoon season the vertical upward transport caused by radiative heating at the top the Asian monsoon anticyclone is faster than elsewhere in the tropics (Vogel et al., 2019).

5. *P4L1-2: How to show the size, radius or diameter?*

The reviewer raised a very relevant point here. In the revised version of the paper we give more details about size distribution of ATAL particles as follows:

Only few measurements of size distributions of particles in the ATAL are available, but measurements with an optical particle counter (OPC) (Deshler et al., 2003) from Hyderabad, India and measurements with a Printed Optical Particle Spectrometer (POPS) (Gao et al., 2016) from Kunming, China, (both in 2015) confirm the presence of the ATAL (Yu et al., 2017; Vernier et al., 2018). The OPC measurements indicate that the concentration of ATAL particles is highest close to the cold point tropopause and decline towards greater altitudes; concentrations (up to about 25 particles per $cm^{-3}$) are dominated by particles in the $r > 0.094\,\mu m$ channel, whereas particle number concentrations for the $r > 0.15\,\mu m$ channel, and the $r > 0.30\,\mu m$ channel are lower by a factor of 30 and 300, respectively (Vernier et al., 2018, Fig. 9).

6. *P4L3-4: 0.6W/m2 and 0.5K???*

Many thanks for this important advice. We revised the paragraph as follows in the revised version of the paper:

Based on CALIOP measurements, the summertime aerosol optical depth over Asia associated with the ATAL has increased from $\approx 0.002$ to $0.006$ between 1995 and 2013, resulting in a short-term regional forcing at the top of the atmosphere of $-0.1$ W/m$^2$ – compensating about one third of the comparable radiative forcing associated with the global increase in $CO_2$ (Vernier et al., 2015). The regional radiative forcing caused by the ATAL, differs for clear and total sky conditions; total sky calculations show less shortwave radiative forcing over the monsoon region because of cloudiness (Vernier et al., 2015).

It is likely that, over Asia in the past $\sim 20$ years, the altered radiative forcing has led to summertime reductions in surface temperature, although this effect is not quantified yet. However, the radiative forcing caused by the ATAL could be compared with the global aerosol forcing caused by moderate volcanic eruptions since 2000, which translates into a surface cooling of $0.05$ to $0.12$ K (Ridley et al., 2014).

7. *P4L7-10: In some ATAL studies, volcanic eruptions are removed, because volcanic signal is much stronger than ATAL, which will mask the effect of ATAL.*

Following the reviewer's advice we revised this paragraph in the revised version of the paper as follows:

The co-occurrence of the ATAL with volcanic eruptions in the region further enhances the radiative forcing by the ATAL; Fairlie et al. (2014, Fig. 8), based on monthly accumulations of CALIOP aerosol data between 14 and 40 km altitude, reported a top of the atmosphere radiative forcing locally over Asia and Europe during July 2011 in response to the Nabro volcanic eruption of about $-0.8$ to $-1.5\,\mathrm{W/m^2}$. Therefore, periods of volcanic eruptions are removed in some ATAL studies; because the volcanic signal is much stronger than the ATAL signal itself, a volcanic signal will mask the ATAL (e.g. Thomason and Vernier, 2013; Vernier et al., 2015).

The impact of the ATAL is furthermore modulated by El Niño. During El Niño, the ATAL is thicker and broader over the Indian region, resulting in a reduction of the solar flux and a surface cooling of about 1 K over North India. An elevated ATAL over South Asia exacerbates the severity of Indian droughts (Fadnavis et al., 2019).

8. *P4L28-29: Nitrate aerosol is dominant in the ATAL is first suggested by Gu et al. (2016) by simulation. Gu, Y., H. Liao, and J. Bian, 2016: Summertime nitrate aerosol in the upper troposphere and lower stratosphere over the Tibetan Plateau and the South Asian summer mon- soon region, Atmos. Chem. Phys., 16, 6641-6663, doi:10.5194/acp-16-6641-2016.*

Many thanks for this advice. We revised the paragraph as follows in the revised version of the paper:

Only limited information on the chemical composition of the ATAL particles is available from measurements so far. From simulations, there is evidence that desert dust is lifted to UTLS altitudes and entrained into the ATAL (Fadnavis et al., 2013; Lau et al., 2018; Yuan et al., 2019). Further, simulations show elevated aerosol concentrations of sulphate, nitrate, ammonium, black carbon and organic carbon within the ATAL with contributions of nitrate dominating (Gu et al., 2016; Fairlie et al., 2020). Moreover,

Fairlie et al. (2020) found in model simulations for summer 2013 a different chemical compositions of the ATAL depending on the location within the Asian monsoon anticyclone with nitrate aerosol as the dominant component on the southern flank of the anticyclone.

9. *P6L30: CI>7.0, BSR940>2 AND Sice>70% → OR*

   We have replaced the following text in the ACPD version:

   'We reject layers with CI > 7.0, $BSR_{940} \geq 2$ and $S_{ice} > 70\%$ as cirrus clouds (Vernier et al., 2015; Li et al., 2018; Brunamonti et al., 2018)....'

   in the revised version of the paper with:

   We reject layers showing a CI > 7.0, and a $BSR_{940} \geq 2$ as well as an enhanced ice saturation (Sice > 70%) as cirrus clouds; clearly the CI criterion matters most for the detection of cirrus particles (Vernier et al., 2015; Li et al., 2018; Brunamonti et al., 2018).

10. *P6L30: Aerosol layers without cirrus can exist under the condition Sice > 70%.*

    We agree, see text change above item 9.

11. *P6L32: Could you provide the vertical range of UTLS?*

    We replaced 'UTLS' by the altitude range as follows:

    Other sections of the profiles measured during the August soundings, which show substantially elevated values of $BSR_{455}$ between ∼140 and 70 hPa (≈ 14–18 km), are classified as ATAL.

12. *P9L11-14: This 'somewhat different picture' can be explained by the results from Pan et al. (2014) as mentioned above. The distribution of lapse-rate minimum levels is compact in potential temperature scale but diffuse in the*

*altitude scale.*

In Pan et al. (2014), it is described that the distribution of the level of minimum stability (LMS) is more compact in potential temperature space and that the cold point tropopause (CPT) levels are more compact in the altitude space. This is consistent with the behaviour of temperature and potential temperature profiles in these two regions – LMS is the region of near constant potential temperature so the minimum gradient in this region can potentially vary over a large altitude range. The CPT has a very well defined altitude from the temperature profiles, but it is a region where potential temperature has a large gradient. However, in our work the distributions of ATAL with pressure and potential temperature as the vertical coordinate are discussed. It is difficult to directly link the distributions of the levels of CPT and LMS to the ATAL.

We revised this sentence as follows:

Using the backscatter measurements with potential temperature as the altitude scale, a somewhat different picture emerges than when using pressure as the altitude scale because the potential temperature has a large gradient in this altitude region.

13. *10L1: This result can also explained by the results on CPT from Pan et al. (2014).*

See above item 12.

14. *P18L1: The two branches are not easily found in the figure, could you show more clearly?*

We revised the description of the two branches to make the difference more clear in revised version of the manuscript as follows:

Focusing on the trajectories from the BL (Fig. 8; in the ACPD Version) two branches, a western and eastern branch, of trajectories are found depending on the altitude of the measurements (Fig. 7a; in the ACPD Version); trajectories from above 370 K are from eastern and trajectories from below 370 K

are from western longitudes. In the western branch of the trajectories the air masses are transported around the western mode of Asian monsoon anticyclone, while air masses from the eastern branch are coming from the Pacific ocean and are transported along the southern edge of the eastern mode of the anticyclone to the measurement location in Nainital.

15. *P27L1-6: Why post-monsoon cases are used as background signal for aerosols in the UTLS? Yes, theres no ATAL during winter. But, the general circulation is quite different for summer and winter, so obviously the source regions are different. Could you use cases with no ATAL during summer as background signal?*

We agree that the general circulation in winter is very different to summer conditions and therefore the source regions are different for air parcels in the UTLS over Asia during summer and winter. Therefore the enhancement in backscatter ratio ($BSR_{455}$) in August compared to the November measurements is a good measure to infer the strength of the ATAL in summer in relation to conditions in winter, when more unpolluted marine air masses are transported to the UTLS over Asia. Therefore we use the November cases as background signal.

It would be also interesting to use 'no ATAL cases' during summer as background for further analysis however our study is focused on the COBALD measurements in Nainital 2016 which is a small statistical data base of 15 valid flights in August and two during November. In this data base, there are only two 'no ATAL cases' during summer both overlaid by a cirrus cloud. Therefore, we consider it premature using these two 'no ATAL cases' as the background signal.

16. *P27: For case 2, beside source regions, other parameters impacting ATAL such as temperature and Sice should also be considered.*

We agree and added the following sentence:

However, we can not exclude that also aerosol removal processes depending on temperature and $S_{ice}$ such as in-situ cirrus formation contribute to the

17. *P28L26: Some in situ measurements show that the aerosols concentration in the middle troposphere is very low, which should be considered in the argument.*

We agree that the aerosols concentration in the middle troposphere is very low in some in situ measurements which is also visible in the measurements in Nainital presented in this work (see Fig. 2 in the ACPD version). In the paper, we argue that the uplift (convection) of air masses from the lower troposphere could transport enhanced concentrations of aerosol particles to ATAL altitudes which is independent from the low aerosol concentration in the middle troposphere. This is indeed consistent with the OPC measurements presented by Vernier et al. (2018) that found much lower aerosol concentrations in the middle troposphere that at the ground or in the ATAL. We have now mentioned this point in the paper explicitly.

We have replaced the text in the ACPD version:

'...however the uplift (convection) of air masses from the lower troposphere could transport enhanced concentrations of aerosol particles to ATAL altitudes (as proposed by Vernier et al. (2018). Vernier et al. (2018) found in some balloon-borne measurements of the ATAL from Hyderabad, India, in summer 2015 that the presence of cirrus is associated ...'

in the revised version of the paper with:

...however the uplift (convection) of air masses from the lower troposphere could transport enhanced concentrations of aerosol particles from close to the ground to ATAL altitudes without much influence of aerosol in the middle troposphere (see Fig. 2 in the ACPD). This concept is consistent with the results of Vernier et al. (2018), who found in OPC balloon-borne measurements of the ATAL from Hyderabad, India, in summer 2015 that aerosol concentrations and in the ATAL are of the same magnitude, while much lower aerosol concentrations prevail in the middle troposphere. In their balloon measurements Vernier et al. (2018) further found that the presence of cirrus is associated ...

18. *P31L24-26: Factors impacting the variability of the ATAL include not only source regions, but also other parameters related the formation and growth of aerosol, the latter should also be considered.*

We agree and added the reviewer's argument to the revised version of the manuscript as follows:

In this work, we show that there is a variety of factors impacting the variability of the ATAL: continental convection, tropical cyclones (maritime convection), dynamics of the anticyclone and stratospheric intrusions. All these factors contribute to the observed day-to-day variability of the ATAL found over Nainital in August 2016. The ATAL is also impacted by other processes related to the formation and growth of aerosol as well as by aerosol removal processes such as in-situ cirrus formation.

and add the following text to the second paragraph of the conclusions:

... However, we have only one no ATAL measurement in August 2016 (partly influenced by cirrus), therefore we can not exclude that also aerosol removal processes such as in-situ cirrus formation contribute to the existence of no ATAL measurements. More balloon-borne measurements would be required to validate this hypothesis.

19. *P32L1-4: Possibly, satellite data for aerosol can be used.*

Many thanks for the reviewer's suggestion, however the strong day-to-day variability of the altitude, the vertical extent and the backscatter intensity of the ATAL at UTLS altitudes is not visible in the climatological mean values of the ATAL derived by satellite observations. Therefore we are not sure if satellite data could really help to prove our hypothesis that the main driver for 'no ATAL cases' are the dynamics of the Asian monsoon anticyclone. In any event, an in-depth study of the averaging procedure necessary to derive ATAL information from satellite observations would be required for such an analysis. This is more than what we can achieve in this paper. Furthermore, a larger data base of high-resolution measurements (e.g. from balloons or high-altitude aircraft) would also be very helpful to get a deeper understanding of 'no ATAL cases' during Asian summer monsoon.

20. *P32L20-22: I think its too early to talk about the regulations, because we still dont know whether the ATAL existence is good or bad to human beings.*

We agree and discuss this issue more controversially in the revised version of the paper by adding the following sentence:

[revised manuscript text omitted]

---

## Referee Comment (RC2) · Anonymous Referee #2 · 7 Aug 2020

Review of "Strong Variability of the Asian Tropopause Aerosol Layer at the foothills of the Himalayan" by Hanumanthu et al., 2020.

Understanding the nature, origin and impacts of the Asian Tropopause Aerosol Layer has been a research focus for nearly a decade. Recent airborne campaigns conducted in Asia during the Summer Monsoons have provided a wealth of information about the ATAL that are rapidly advancing our understanding of this phenomenon. As a part of the StratoClim field experiment that took place in Nepal and India in 2017, this study present results from the balloon flights conducted from Nainatal in August 2017 compared with those obtained in November 2016. The balloon flights reveal an impor-

tant day-to-day variability of the Scattering Ratio (SR) taken as a relative measure of aerosol loadings. In order to understand the causes of those fluctuations, the authors run the CLAMS trajectory model to distinguish the origin of air masses in the Boundary Layer, the free troposphere and the lower stratosphere from different part of Asia. They concluded that large SR values within the ATAL tend to be associated with air masses from the Tibetan plateau, Himalayan foothills and lands while oceanic origin tend to result in a depletion of UTLS aerosols. Despite some grammatical mistakes and relatively lengthy manuscript, which could be, shorten and better summarize, this is an interesting study, which merits its publication in ACP. However, I suggest significant revisions to make this possible. Because deep convection is a fundamental transport pathways for air mass to move from the Boundary Layer into the Upper Troposphere and Lower Stratosphere during the Monsoon, the coarse resolution of the meteorological field used to run CLAMS likely result in misrepresentation of the vertical transport pathways especially after a few days when the likelihood of encountering deep convection is very high. The manuscript lacks a deeper analysis of the role of convective storms that influence the vertical transport of air masses and those measurements. Other studies have used Cloud Top Temperature as a proxy for deep convection to find out the location where air masses are influenced by deep convection and I believe that this study would need to adopt a similar approach to be more convincing. Below are additional technical comments of this paper that the authors may want to consider:

1) Title. I'm not sure the title translate very well the topics of this paper. Moreover, the term "strong variability" is confusing if not related to time information (in this case day-to-day or intraseasonal variability). 2) P1/L3. I believe that "inside" is not required. It's understood from the previous part of the sentence. 3) P1/L7. "COBALD" does not need to be repeated here. It could be replaced by "compared to those obtained..." 4) P1/L12-L13. Not "composition" but scattering ratio. 5) P1/L18-21. This is related to the major comment I have on this study. How realistic is the vertical transport pathway described here relative to direct injection by deep convection ? 6) P2/L5. Chinese emissions have decreased drastically over the past 2 decades so Sulfur emission in Asia

are overall on the decreasing side so this sentence needs to be modified. 7) P2/L9-L11. I would argue that Chinese balloon-borne measurements and ground-based lidar suggested the presence of aerosol layers over the Tibetan Plateau earlier than satellite observations but the extension of the ATAL was indeed discovered through global satellite observations. Refs : Kim, Y.-S., T. Shibata, Y. Iwasaka, G. Shi, X. Zhou, K. Tamura, and T. Ohashi, 2003: Enhancements of aerosols near the cold tropopause in summer over Tibetan Plateau: Lidar and balloonborne measurements in 1999 at Lhasa, Tibet, China. Lidar Remote Sensing for Industry and Environment Monitoring III, U. N. Singh, T. Itabe, and Z. Liu, Eds., Society of Photo-Optical Instrumentation Engineers (SPIE Proceedings, Vol. 4893), 496–503, https://doi .org/10.1117/12.466090. Tobo, Y., Y. Iwasaka, G.-Y. Shi, Y.-S. Kim, T. Ohashi, K. Tamura, and D. Zhang, 2007: Balloon-borne observations of high aerosol concentrations near the summertime tropopause over the Tibetan Plateau. Atmos. Res., 84, 233–241

8) P2/L17. This sentence needs to be more accurately stated. The paper did not suggest the presence of the ATAL in the 90's but the presence of ammonium nitrate and since we do not know the overall contribution of AN within the ATAL, It's hard to formalize a general statement such as the one here. I suggest being more accurate. 9) P2/L26-27. I would suggest targeting the citations that are most appropriate for this statement. 10) P3/L12. I would suggest being quantitative in this sentence. What are the contributions from India and China? 11) P3/L21. Could you explain why the results seem to be consistent with Brunamunti et al., 2018 ? 12) P4/L4. Is there a reference for those estimates? 13) Overall, the introduction could be improved by organizing the different paragraph with titles. 14) P5/L9. A reference to Pandit et al., 2015 could be added here . Ref: Pandit, A. K., Gadhavi, H. S., Venkat Ratnam, M., Raghunath, K., Rao, S. V. B., and Jayaraman, A.: Long-term trend analysis and climatology of tropical cirrus clouds using 16 years of lidar data set over Southern India, Atmos. Chem. Phys., 15, 13833–13848, https://doi.org/10.5194/acp-15-13833-2015, 2015. 15) P6/L4. A calibration adjustment is needed to fit the COBALD raw signal to the molecular scattering. A few lines describing a little better the procedure

could be added here. 16) P6L16. If I'm not mistaken IST=UTC+5h30...(not 6h). 17) P6.L30. You probably mean to say "we identify ice clouds with...." 18) P9L14. Aerosol scavenging also depends on aerosol size and composition, which affect their ability to uptake water. 19) P13L19. How trustable are trajectories run beyond a week ? 20) P14. Table 2 needs to be better explained. What's the definition of the variable in the table? Residence time in a given layer relative to the sum? 21) P30. Figure 15. Why do you choose to take the mean value? I would suggest to plot the same with the value corresponding to the altitude where the model was initialized 22) Figure 15. What are the correlation coefficient values? I'm not sure if you can draw much conclusions from this plot apart overall tendency. 23) P31/L17. This phrase needs to be nuanced. While it is true that the signal-to-noise ratio from the CALIPSO space-borne lidar does not allow studying day-to-day variability of the ATAL, observations from SAGE II/SAGEIII can be potentially used for that. 24) P31/L24. I don't think the impact of convection, which is not well represented in ERA-Interim, has been fully explored and thus must bias most of the trajectory results presented in this paper.

―――――――――――――――――――

---

## Author Comment (AC2) · 3 Sep 2020

**Author Comment to Referee #2**

**ACP Discussions doi: 10.5194/acp-2020-552-RC1, (Editor - Gabriele Stiller), 'Strong variability of the Asian Tropopause Aerosol Layer (ATAL) in August 2016 at the Himalayan foothills' by Sreeharsha Hanumanthu et al.**
* * *
We thank Referee #2 for important further guidance on how to revise our manuscript. Our reply to the reviewer comments is listed in detail below. Questions and comments of the referee are shown in italics. Passages from the revised version of the manuscript are shown in blue.

*Understanding the nature, origin and impacts of the Asian Tropopause Aerosol Layer has been a research focus for nearly a decade. Recent airborne campaigns conducted in Asia during the Summer Monsoons have provided a wealth of information about the ATAL that are rapidly advancing our understanding of this phenomenon. As a part of the StratoClim field experiment that took place in Nepal and India in 2017, this study present results from the balloon flights conducted from Nainatal in August 2017 compared with those obtained in November 2016. The balloon flights reveal an imporant day-to-day variability of the Scattering Ratio (SR) taken as a relative measure of aerosol loadings. In order to understand the causes of those fluctuations, the authors run the CLAMS trajectory model to distinguish the origin of air masses in the Boundary Layer, the free troposphere and the lower stratosphere from different part of Asia. They concluded that large SR values within the ATAL tend to be associated with air masses from the Tibetan plateau, Himalayan foothills and lands while oceanic origin tend to result in a depletion of UTLS aerosols.*

*Despite some grammatical mistakes and relatively lengthy manuscript, which could be, shorten and better summarize, this is an interesting study, which merits its publication in ACP. However, I suggest significant revisions to make this possible. Because deep convection is a fundamental transport pathways for air mass to move from the Boundary Layer into the Upper Troposphere and Lower Stratosphere during the Monsoon, the coarse resolution of the meteorological field used to run CLAMS likely result in misrepresentation of the vertical transport pathways*

*especially after a few days when the likelihood of encountering deep convection is very high. The manuscript lacks a deeper analysis of the role of convective storms that influence the vertical transport of air masses and those measurements. Other studies have used Cloud Top Temperature as a proxy for deep convection to find out the location where air masses are influenced by deep convection and I believe that this study would need to adopt a similar approach to be more convincing. Below are additional technical comments of this paper that the authors may want to consider.*

The reviewer's major comment regarding the representation of convection within CLaMS back-trajectory calculations driven by ERA-Interim reanalysis is discussed in detail under item # 5. We agree that the manuscript is somewhat long therefore we followed the reviewer's advice and removed Fig. 14b, c as well as Fig. 15b (ACPD version) and the corresponding text in Sect. 4.4. Further, the manuscript was carefully proof-read regarding to grammatical mistakes by several of the authors. To avoid any misunderstanding we would like to point out that the paper contains only balloon-borne measurements from India (Nainital) in 2016. Balloon-borne measurements were also performed in Nepal in summer 2017 in the frame of the StratoClim project, however no COBALD measurements are available for 2017.

**Minor issues:**

1. *Title. Im not sure the title translate very well the topics of this paper. Moreover, the term strong variability is confusing if not related to time information (in this case day-to-day or intraseasonal variability).*

   We agree and changed the title from

   'Strong variability of the Asian Tropopause Aerosol Layer (ATAL) in August 2016 at the Himalayan foothills' to

   Strong day-to-day variability of the Asian Tropopause Aerosol Layer (ATAL) in August 2016 at the Himalayan foothills

2. *P1/L3. I believe that 'inside' is not required. Its understood from the previous part of the sentence.*

We prefer to keep 'inside' to avoid any misunderstanding.

3. *P1/L7. 'COBALD' does not need to be repeated here. It could be replaced by 'compared to those obtained...'*

We prefer to keep 'COBALD' to have a clear message.

4. *P1/L12-L13. Not 'composition' but scattering ratio.*

Thanks for the comment. We revised the sentence

'We identify the transport pathways of air parcels contributing to the ATAL over Nainital in August 2016, as well as the source regions of the air masses contributing to the composition of the ATAL.'

in the revised version of the paper as follows:

We identify the transport pathways as well as the source regions of air parcels contributing to the ATAL over Nainital in August 2016.

5. *P1/L18-21. This is related to the major comment I have on this study. How realistic is the vertical transport pathway described here relative to direct injection by deep convection ?*

Within the Nainital measurements in summer 2016 the ATAL is located from 360 K up to 420 K. The top of the convective outflow level is around 360 K. Deep convection events are most likely underestimated in ERA-Interim. The new high-resolution ECMWFs next-generation reanalysis ERA5 (Hoffmann et al., 2019; Hersbach et al., 2020) has much higher spatial and temporal resolution than ERA-Interim. A recent comparison between CLaMS trajectories driven by ERA-Interim and by ERA5 investigated the impact of tropical cyclones on ozone and water vapour measured during the SWOP balloon-campaign in 2009 and 2015 in Kunming (China) within the Asian monsoon anticyclone (Li et al., 2020). Different vertical transport via deep convection is found depending on the employed reanalysis data

(ERA-Interim, ERA5) and vertical velocities (diabatic, kinematic). Both the kinematic and the diabatic trajectory calculations using ERA5 data show faster and stronger vertical transport than ERA-Interim primarily due to ERA5's better spatial and temporal resolution, likely resolving more convective events. Although the details of the vertical transport are different in all the cases studied by Li et al. (2020), the convective upward transport by tropical cyclones is found in ERA-Interim (diabatic) as well as in ERA5 (diabatic and kinematic). The location of the convective updraft is compared with brightness temperature from IR channel of FY-2D satellite and with cloud top temperature from FY-2G satellite showing that ERA-Interim (diabatic) as well as ERA5 (diabatic, kinematic) trajectories have the convective upward transport in the region of the tropical cyclone, whereby ERA5 (diabatic) fits best with the center of the cyclone. Further, the transport pathway above the convective outflow level driven by both the anticyclonic flow and diabatic heating in the region of the Asian monsoon anticyclone ('upward spiralling range') is found in both reanalysis data (ERA-Interim, ERA5). Li et al. (2020) show that low ozone and low water vapour mixing ratios near the tropopause measured in August 2009 and 2015 in Kunming are the result of the interplay between the uplift of dry ozone-poor maritime air within tropical cyclones and the transport in the UTLS driven by the Asian monsoon anticyclone.

We are aware that using ERA5 for CLaMS trajectory calculations for the ATAL measurements in Nainital 2016 would yield deeper insights into the impact of deep convection to ATAL, however a study using ERA5 goes beyond the scope of the study presented here. Further studies are prerequisite to validate convection and diabatic heating rates in ERA5. In this study, we highlight the day-to-day variability of ATAL during August 2016 and its relation to continental and maritime convection. The use of high-resolution meteorological data and the use of satellite measurements (e.g. Cloud Top Temperature) would be necessary to identify the detailed location of single convection events. However, the results by Li et al. (2020) encourage us that the representation of convection in ERA-Interim is adequate for the study presented here.

6. *P2/L5. Chinese emissions have decreased drastically over the past 2 decades so Sulfur emission in Asia are overall on the decreasing side so this sentence needs to be modified.*

We agree with the reviewer's comment and revised the sentence

'Because of the strong growth of Asian economies, increasing anthropogenic emissions in the future are expected to enhance the thickness and intensity of the ATAL, thereby also enhancing the global stratospheric aerosol loading, which likely impacts surface climate.'

in the revised version of the paper as follows:

On the one hand increasing anthropogenic emissions in the future are expected due to the strong growth of Asian economies, on the other hand implementation of new emission control measures (in particular in China) have reduced substantially the anthropogenic emissions of some pollutants contributing to the ATAL. It needs to be monitored in the future, whether the thickness and intensity of the ATAL will further increase, which likely impacts surface climate.

Further, Chinese emissions are discussed in the Introduction of the ACPD version as well (P3, L25-27).

7. *P2/L9- L11. I would argue that Chinese balloon-borne measurements and ground-based lidar suggested the presence of aerosol layers over the Tibetan Plateau earlier than satellite observations but the extension of the ATAL was indeed discovered through global satellite observations. Refs.:*

*Kim, Y.-S., T. Shibata, Y. Iwasaka, G. Shi, X. Zhou, K. Tamura, and T. Ohashi, 2003: Enhancements of aerosols near the cold tropopause in summer over Tibetan Plateau: Lidar and balloonborne measurements in 1999 at Lhasa, Tibet, China. Lidar Remote Sensing for Industry and Environment Monitoring III, U. N. Singh, T. Itabe, and Z. Liu, Eds., Society of Photo-Optical Instrumentation Engineers (SPIE Proceedings, Vol. 4893), 496503, https://doi.org/10.1117/12.466090.*

*Tobo, Y., Y. Iwasaka, G.-Y. Shi, Y.-S. Kim, T. Ohashi, K. Tamura, and D. Zhang, 2007: Balloon- borne observations of high aerosol concentrations near the summertime tropopause over the Tibetan Plateau. Atmos. Res., 84, 233241*

We thank the reviewer for these references; we were indeed not aware of these measurements. As suggested the measurements in August 1999 are now mentioned in the paper, when the ATAL is introduced. We added the following text:

Although the large horizontal extent of the ATAL was first seen in the CALIOP measurements (Vernier et al., 2011, 2015), first observations of enhanced number concentrations of sub-micron aerosol particles between 130-70 hPa in the Asian summer monsoon were made during a balloon ascent from Lhasa (29.7°N, 91.1°E) already in August 1999 (Kim et al., 2003; Tobo et al., 2007).

And we are now also referring to the papers in question in the discussion of measurements of size distributions of particles in the ATAL in the introduction further below:

Only few measurements of size distributions of particles in the ATAL are available. Early size-resolved measurements using a balloon-borne optical particle counter (OPC) in August 1999 showed high number concentrations (0.7–0.8 particles $cm^{-3}$) of aerosol particles with radii of 0.15–0.6 $\mu$m between about 130–70 hPa in the Asian summer monsoon (Kim et al., 2003; Tobo et al., 2007).

8. *P2/L17. This sentence needs to be more accurately stated. The paper did not suggest the presence of the ATAL in the 90s but the presence of ammonium nitrate and since we do not know the overall contribution of AN within the ATAL, It's hard to formalize a general statement such as the one here. I suggest being more accurate.*

The paper by Höpfner et al. (2019) contains the following statement: "The spatially resolved AN observations with the CRISTA satellite reveal that enhanced concentrations of AN (0.05-0.3 $\mu$g m$^3$) are located only within the AMA (Fig. 1). These observations between 8 and 16 August 1997 indicate that an ATAL was present in the Asian monsoon UT in summer 1997, years earlier than hitherto thought".

However, we agree that a more careful wording is appropriate here; we have changed the part of the sentence starting with "although" by:

However, Höpfner et al. (2019) reported that as early as 1997, during the Asian monsoon period, enhanced concentrations of solid ammonium nitrate

particles were present throughout the Asian monsoon anticyclone.

9. *P2/L26-27. I would suggest targeting the citations that are most appropriate for this statement.*

Many thanks for this comment. We agree that a plenty of citations are included for this statement. However, in these citations different measurements and chemical species are used. Therefore, we think it is useful to cite all of them here.

10. *P3/L12. I would suggest being quantitative in this sentence. What are the contributions from India and China?*

We agree and added the percentages from India and China to the revised version of the paper as follows:

Source apportionment of the model simulations by Fairlie et al. (2020) indicated the dominance of the contribution of regional anthropogenic emissions from China and the Indian subcontinent (both $\sim$ 30% attributable to regional $SO_2$ sources) to aerosol concentrations in the ATAL in August 2013.

In addition, the Chinese $SO_2$ emissions (Zheng et al., 2018) are also discussed in the Introduction of the ACPD version (P3, L25-27).

11. *P3/L21. Could you explain why the results seem to be consistent with Brunamunti et al., 2018 ?*

Brunamonti et al. (2018) denote the altitude range between maximum convective outflow and the cold point tropopause as the Asian tropopause transition layer (ATTL). Above, the ATTL, based on $H_2O$ measurements and trajectory calculations, Brunamonti et al. (2018) found a layer of a confined air in the lower stratosphere.

Vogel et al. (2019) denote the region where air masses are uplifted by diabatic heating across the (lapse rate) tropopause from about 360 K up to 460 K as the 'upward spiralling range'. The higher the air masses are above the thermal tropopause, the larger the contribution of air masses is from

outside the Asian monsoon anticyclone from the stratospheric background coming into the upward spiralling flow.

Both concepts explain the confinement of air in the Asian monsoon anticyclone directly above the cold point tropopause. The higher the air masses are above the thermal tropopause, the weaker is the confinement that can be explained by the larger contribution of air masses from the stratospheric background. This yields a vertical profile of ATAL as shown in Fig. 1 (right; in the ACPD Version of the paper). In Fig. 1 (right), the potential temperature is considered as the vertical coordinate, a top of the ATAL is not clearly defined. Only a slow decrease of averaged $BSR_{455}$ with altitude is observed. This observation is consistent with a decreasing confinement of the air mass within the Asian monsoon anticyclone with increasing potential temperature (Brunamonti et al., 2018; Vogel et al., 2019).

12. *P4/L4. Is there a reference for those estimates?*

Many thanks for this important advice. We revised the paragraph as follows in the revised version of the paper:

Based on CALIOP measurements, the summertime aerosol optical depth over Asia associated with the ATAL has increased from $\approx 0.002$ to $0.006$ between 1995 and 2013, resulting in a short-term regional forcing at the top of the atmosphere of $-0.1$ W/m$^2$ – compensating about one third of the comparable radiative forcing associated with the global increase in $CO_2$ (Vernier et al., 2015). The regional radiative forcing caused by the ATAL, differs for clear and total sky conditions; total sky calculations show less shortwave radiative forcing over the monsoon region because of cloudiness (Vernier et al., 2015).

It is likely that, over Asia in the past $\sim 20$ years, the altered radiative forcing has led to summertime reductions in surface temperature, although this effect is not quantified yet. However, the radiative forcing caused by the ATAL could be compared with the global aerosol forcing caused by moderate volcanic eruptions since 2000, which translates into a surface cooling of $0.05$ to $0.12$ K (Ridley et al., 2014).

13. *Overall, the introduction could be improved by organizing the different*

*paragraph with titles.*

Caused by the comments of Reviewer #1 and #2 we carefully revised the introducing in several places. Therefore, the introduction in the revised version is somewhat longer than the ACPD version in contrast to the Reviewer's advice to shorten the manuscript. We grouped related topics in different paragraphs without using subtitles.

14. *P5/L9. A reference to Pandit et al., 2015 could be added here . Ref: Pandit, A. K., Gadhavi, H. S., Venkat Ratnam, M., Raghunath, K., Rao, S. V. B., and Jayaraman, A.: Long-term trend analysis and climatology of tropical cirrus clouds using 16 years of lidar data set over Southern India, Atmos. Chem. Phys., 15, 1383313848, https://doi.org/10.5194/acp-15-13833- 2015, 2015.*

    We agree and have added Pandit et al. (2015) at P5/L9 (ACPD Version).

15. *P6/L4. A calibration adjustment is needed to fit the COBALD raw signal to the molecular scattering. A few lines describing a little better the procedure could be added here.*

    Many thanks for this comment. We have replaced the following text in the ACPD version

[revised manuscript text omitted]

20. *P14. Table 2 needs to be better explained. What's the definition of the variable in the table? Residence time in a given layer relative to the sum?*

We added in the revised version of the paper the following lines to the caption of Fig. 2:

In the second and third row, the selection criteria for BL, LT, UT, and LS are listed. Trajectories are considered ending in the BL, when they are located for the first time below about 2–3 km (i.e., hybrid coordinate $\zeta \leq 120$ K). The location of this point is referred to as 'end point' of the trajectory in the model boundary layer. For the remaining trajectories ending at atmospheric altitudes ($\zeta > 120$ K), a potential temperature criterion ($\Theta$) is employed to discriminate between origins in LT, UT and LS.

The parameters within Tab. 2 are also explained within Sect. 4 (P13 L29 – P15 L3; ACPD Version).

21. *P30. Figure 15. Why do you choose to take the mean value? I would suggest to plot the same with the value corresponding to the altitude where the model was initialized.*

It seems that there is a misunderstanding. In Fig. 15 (ACPD version) the mean value of the backscatter intensity ($\overline{\mathrm{BSR}}_{455}$) between bottom and top of ATAL is shown calculated for each balloon sounding using the binned data (see Tab. 1; ACPD version) as explained in Sect. 3.1. Cirrus clouds between top and bottom are excluded in the calculation of $\overline{\mathrm{BSR}}_{455}$. Therefore, $\overline{\mathrm{BSR}}_{455}$ is the backscatter intensity averaged over the ATAL altitude range listed in Tab. 1 for each flight.

We added the following sentence to the figure caption of Fig. 15 (ACPD version) to the revised version of the paper

$\overline{\mathrm{BSR}}_{455}$ is the backscatter intensity averaged over the ATAL altitude range listed in Tab. 1 for each flight (details see Sect. 3.1; ACPD version).

22. *Figure 15. What are the correlation coefficient values? Im not sure if you can draw much conclusions from this plot apart overall tendency.*

Many thanks for this comment. We added a new table (Tab. 1 within this authors' reply; Tab. 5 in the revised version) showing the correlation coefficients for each region and added the following paragraph to the revised version of the manuscript:

[revised manuscript text omitted]

---

## Author Response (AR2)

**Letter to the Editor**

**ACP Discussions doi: 10.5194/acp-2020-552-RC1, 'Strong variability of the Asian Tropopause Aerosol Layer (ATAL) in August 2016 at the Himalayan foothills' by Sreeharsha Hanumanthu et al.**

Dear Gabi,

many thanks for handling our manuscript and the good news that our paper is accepted. We prepared the final version of our manuscript for ACP. We corrected a typo in the figure caption of Fig. 9 and 12 ('2019' → '2016') and updated the reference Jorge et al, 2020 ('submitted' → AMTD).

Best wishes

Bärbel